# ComPhy: Composing Physical Models with end-to-end Alignment

**Alessandro Trenta,**[*] **Andrea Cossu, Davide Bacciu**
Department of Computer Science
University of Pisa, Italy
`alessandro.trenta@phd.unipi.it,`
`andrea.cossu@unipi.it, davide.bacciu@unipi.it`

## Abstract

Real-world phenomena typically involve multiple, interwoven dynamics that can be elegantly captured by systems of Partial Differential Equations (PDEs). However, accurately solving such systems remains a challenge. In this paper, we introduce ComPhy (CP), a novel modular framework designed to leverage the inherent physical structure of the problem to solve systems of PDEs. CP assigns each PDE to a dedicated learning module, each capable of incorporating state-of-the-art methodologies such as Physics-Informed Neural Networks or Neural Conservation Laws. Crucially, CP introduces an end-to-end alignment mechanism, explicitly designed around the physical interplay of shared variables, enabling knowledge transfer between modules, and promoting solutions that are the result of the collective effort of all modules. CP is the first approach specifically designed to tackle systems of PDEs, and our results show that it outperforms state-of-the-art approaches where a single model is trained on all PDEs at once.

## 1 Introduction

Machine Learning is rapidly developing as a new tool to solve dynamical and physical systems, particularly for Partial Differential Equations (PDEs). These methodologies have applications in real-world problems, including weather prediction (Pathak et al., 2022), fluid modeling (Zhang et al., 2024), quantum mechanics (Mo et al., 2022), and molecular dynamics (Behler & Parrinello, 2007). While standard supervised methods require a classical solver or real sensors to gather data and train the model, unsupervised approaches can solve systems of PDEs solely from their definition. Physics-Informed Neural Networks (PINNs, Raissi et al. (2019)) are considered one of the most promising approaches for physical and scientific applications. Neural Conservation Laws (NCL, (Richter-Powell et al., 2022)) represent an alternative approach that guarantees a solution with zero divergence with respect to the inputs. We formally introduce PINN and NCL in Section 2.2.

When considering *systems* of PDEs, current approaches add a loss term for each PDE. However, this approach can be unstable and produce unwanted results. For example, PINNs often fail to converge (Krishnapriyan et al., 2021; Wang et al., 2022) or to capture the turbulent effects that characterize fluids (Sun et al., 2020; Xiang et al., 2022). Some workarounds require reweighting the loss at each epoch (Wang et al., 2021; Xiang et al., 2022; McClenny & Braga-Neto, 2023), enforcing boundary conditions (Sun et al., 2020), or dividing the spatial and temporal domains into subdomains (Jagtap et al., 2020; Kharazmi et al., 2021; Meng et al., 2020; Krishnapriyan et al., 2021). However, most of these approaches are not specifically designed for systems of PDEs, or only focus on a specific one.

In this work, we introduce ComPhy (CP), a multi-module approach to solve systems of PDEs by i) allocating one learning module for each PDE and ii) leveraging the physical structure of the problem to *align* modules that learn the same physical variable(s). CP can be applied to any system of PDEs, and the learning modules can be chosen among any of the existing approaches (we experimented with PINNs and NCL). Intuitively, each CP module solves a simpler problem than the overall system, while our end-to-end alignment process integrates the knowledge from all modules to devise a collective

---

[*]Corresponding author.

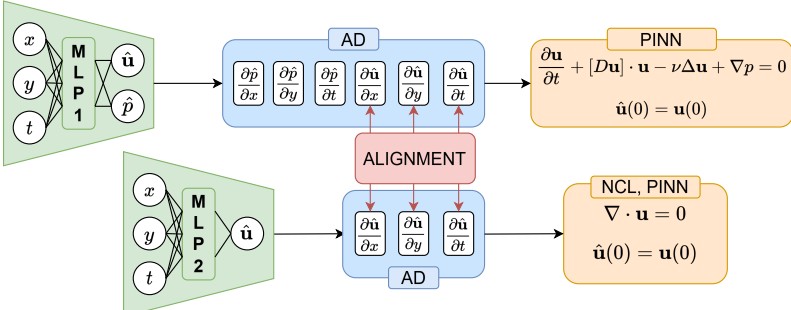

Figure 1: Visual representation of ComPhy for the Navier-Stokes equations. Each module (in this case, PINN or NCL) learns one PDE. The derivatives of each module, computed with Automatic Differentiation (AD) (Baydin et al., 2018), are used to optimize each PDE and to align the other module. CP only uses the first module for inference as it predicts all the variables of the system.

solution. Practically, the alignment encourages the outputs of different modules representing the same physical quantity to be similar. We propose several alignment losses, highlighting the role played by the derivatives in transferring physical knowledge between models (Trenta et al., 2026).

While the training phase involves all learning modules, during inference CP uses the minimum number of modules needed to predict all relevant variables. Most of the time, one module is sufficient. We show that CP achieves state-of-the-art performance, outperforming alternatives where a single learning module is trained on all the equations of the system.

Our contributions are: (i) ComPhy, a new methodology to solve systems of PDEs by allocating one module for each equation while aligning modules that learn the same physical information. (ii) We introduce and evaluate three different alignment approaches. (iii) We evaluate several CP configurations on case studies involving systems of two and three PDEs. We achieve state-of-the-art results in all cases. We analyze the backpropagated gradients through the modules, showing why CP trains better than PINNs. (iv) We evaluate CP on two challenging real-world physical systems with 3 and 5 equations, achieving state-of-the-art results.

## 2 COMPHY

We introduce ComPhy, our end-to-end and modular approach for solving systems of PDEs. Our approach involves: (i) learning each PDE with a single module while (ii) constraining and refining the learning process of each module through shared knowledge and exchange of physical information. In particular, the second point is implemented via an alignment loss between the modules.

### 2.1 METHODOLOGY

To set the stage, let us consider points $\boldsymbol{x}$ sampled from a spatial domain $\Omega \subseteq \mathbb{R}^n$ with time in the interval $[0, T]$. We call $\partial\Omega$ the boundary of the domain $\Omega$. We define a system of $N$ PDEs by

$$\begin{cases} \mathcal{F}_i[\boldsymbol{u}] = 0, & \text{(PDE i) for } i = 1, \ldots, N \\ \mathcal{B}[\boldsymbol{u}] = b(t, \boldsymbol{x}), & \text{(BC)} \\ \boldsymbol{u}(0, \boldsymbol{x}) = g(\boldsymbol{x}), & \text{(IC)} \end{cases} \quad (1)$$

where $\boldsymbol{u}(t, \boldsymbol{x})$ is the solution to the system of $N$ PDEs. A necessary condition for the existence and uniqueness of a solution (Evans, 2022) is to provide the initial condition (IC) $\boldsymbol{u}(0, \boldsymbol{x}) = g(\boldsymbol{x})$, as well as the boundary condition (BC) $\mathcal{B}[\boldsymbol{u}] = b(t, \boldsymbol{x})$. The PDEs must be satisfied for every $(t, \boldsymbol{x}) \in [0, T] \times \Omega$, the IC for every $\boldsymbol{x} \in \Omega$, while the BC for every $(t, \boldsymbol{x}) \in [0, T] \times \partial\Omega$.

State-of-the-art models like PINNs may suffer from optimization problems when multiple PDEs are involved (Sun et al., 2020; Wang et al., 2022), due to the ill-conditioning and the complex loss landscape of PDE residual terms (Krishnapriyan et al., 2021), which can lead to strong imbalances in the gradient norms of the different loss terms (Wang et al., 2022). ComPhy avoids this issue

by using different modules to optimize the different PDEs of the system separately. This makes each optimization problem simpler than solving the whole system, while cross-PDE information is acquired through alignment. In the CP modular architecture, each network takes as input the same coordinates $(t, \boldsymbol{x})$ and predicts the relevant variables in its equation. Modules can sometimes solve more than one equation, but no module solves the whole system. Each module learns the IC, BC, and the PDE(s) of interest through the weighted sum of the respective loss terms:

$$
\begin{aligned}
\mathcal{L}_{\text{module i}} =& \lambda_{\text{BC}}\mathcal{L}_{\text{BC}} + \lambda_{\text{IC}}\mathcal{L}_{\text{IC}} + \lambda_{\text{PDE i}}\mathcal{L}_{\text{PDE i}} \\
=& \lambda_{\text{BC}}\|\hat{\boldsymbol{u}}(t,\boldsymbol{x}) - b(t,\boldsymbol{x})\|^2_{L^2([0,T]\times\partial\Omega)} + \lambda_{\text{IC}}\|\hat{\boldsymbol{u}}(0,\boldsymbol{x}) - g(\boldsymbol{x})\|^2_{L^2(\Omega)} + \lambda_{\text{PDE i}}\mathcal{L}_{\text{PDE i}}.
\end{aligned}
\tag{2}
$$

The loss term $\mathcal{L}_{\text{PDE i}}$ depends on the particular choice of the learning module (Section 2.2). Figure 1 shows the CP architecture for the Navier-Stokes PDEs, which we discuss in Section 2.3.

**Alignment.** If the PDEs in the system are not all satisfied at once, the solution to the system might not be unique (Evans, 2022). Hence, we need to provide each module with information about the state of the others, at least on the parts with the same physical information. This is the role of our *alignment process*.

Let $\boldsymbol{v}$ be the subset of shared variables predicted by two different modules. The alignment loss fosters consistency between the predicted $v \in \boldsymbol{v}$ by both models. We focus on 2 alignment losses acting on the prediction of $\boldsymbol{v}$ made by module $i$ ($\hat{\boldsymbol{v}}_i$), and by module $j$ ($\hat{\boldsymbol{v}}_j$):

$$
\begin{aligned}
\text{SOB: } \mathcal{L}_{\text{align i,j}} &= \|\hat{\boldsymbol{v}}_i - \hat{\boldsymbol{v}}_j\|^2_2 + \|\mathbf{J}\hat{\boldsymbol{v}}_i - \mathbf{J}\hat{\boldsymbol{v}}_j\|^2_2, \\
\text{DERL: } \mathcal{L}_{\text{align i,j}} &= \|\mathbf{J}\hat{\boldsymbol{v}}_i - \mathbf{J}\hat{\boldsymbol{v}}_j\|^2_2,
\end{aligned}
\tag{3}
$$

where $\mathbf{J}\hat{\boldsymbol{v}}_i$ is the Jacobian of the predictions for $\boldsymbol{v}$ of module $i$ with respect to all its inputs. SOB aligns the Sobolev norm $W^2$ (Maz'ya, 2011) while DERL aligns the $L^2$ norm on the derivatives only. Incorporating derivative information improves the convergence (Czarnecki et al., 2017) and imposes a stricter resemblance between the models' outputs. In physical systems, derivatives describe the system's dynamics and provide sufficient information to determine its evolution. Furthermore, they are the most effective target for transferring and distilling physical information between models (Trenta et al., 2026). To substantiate the claim, we also consider the OUTL alignment loss $\|\hat{\boldsymbol{v}}_i - \hat{\boldsymbol{v}}_j\|^2_2$ as well, which is based only on the $L^2$ distance of the outputs. Given our discussion, we expect this to perform much worse than the others. See Appendix B.2 for more details on the alignment losses. The complete loss of the CP model can be expressed as

$$
\mathcal{L}_{\text{CP}} = \lambda_{\text{align}} \sum_{i,j} \mathcal{L}_{\text{align i,j}} + \sum_{i=1}^{N} \mathcal{L}_{\text{module i}},
\tag{4}
$$

where the sum on $i, j$ spans each pair of modules sharing at least one variable. All $L^2$ losses are implemented using the empirical version, that is, the mean squared error on collocation points. In reality, we see that it is sufficient to consider a simplified version of this loss, which includes only the alignment terms between the inference module and the others. For more details, see Appendix C.1.
**Inference.** CP trains all the modules end-to-end. During inference, CP uses only one or a subset of the modules. In particular, it is sufficient to include the minimum number of modules required to predict all the variables present in the solution. This often saves a considerable amount of inference time, as even a single module is often sufficient to predict the output.

## 2.2 LEARNING MODULES

We now introduce the learning modules we adopted for our experiments with CP. We consider existing state-of-the-art models that are widely used to learn physical systems. As we demonstrate in our experimental sections, Sections 3 and 4, optimizing all equations within a single module is suboptimal for learning challenging systems of PDEs due to the different scales and competing objectives of the losses. Our model, based on aligning specialized modules, overcomes these issues and outperforms all baselines.

**Physics Informed Neural Networks (PINNs).** PINNs (Raissi et al., 2019) are among the best-performing and most used models for learning solutions to PDEs. They can be constructed upon

any differentiable architecture. As commonly done in the literature (Raissi et al., 2019; Lau et al., 2024), we consider multi-layer perceptrons (MLPs). PINNs take as input the coordinates of a point $(t, \boldsymbol{x})$ and output the prediction of the solution $\hat{\boldsymbol{u}}(t, \boldsymbol{x})$ to the PDE. To solve the problem, PINNs try to impose the PDE as a soft target for the MLP itself. Partial derivatives of the MLP $\frac{\partial \hat{\boldsymbol{u}}}{\partial \boldsymbol{x}}, \frac{\partial \hat{\boldsymbol{u}}}{\partial t}, \ldots$ are calculated via Automatic Differentiation (AD, Baydin et al. (2018)). Then, the residual on the PDE is calculated using these derivatives $\mathcal{F}[\hat{\boldsymbol{u}}]$. Finally, the $L^2$ loss is used to minimize this residual, together with the IC and BC, obtaining the full loss:

$$\mathcal{L}_{\text{PINN}} = \|\hat{\boldsymbol{u}}(t, \boldsymbol{x}) - b(t, \boldsymbol{x})\|_2^2 + \|\hat{\boldsymbol{u}}(0, \boldsymbol{x}) - g(\boldsymbol{x})\|_2^2 + \|\mathcal{F}[\hat{\boldsymbol{u}}]\|_2^2. \tag{5}$$

**Neural Conservation Laws (NCL).** Being able to output a conservative field by design is the key feature of NCL (Richter-Powell et al. (2022)). To achieve this, the MLP does not directly output the field $\boldsymbol{u}$, but parametrizes an antisymmetric matrix $\mathbf{A}$. Then, the divergence operator $\text{div}(\boldsymbol{f}) = \sum_{i=1}^d \frac{\partial \boldsymbol{f}}{\partial x_i}$ is applied row-wise, obtaining the final vector $\boldsymbol{u}$ with items $\boldsymbol{u}_i = \text{div}(\mathbf{A}_{i\cdot})$. This parameterization ensures that $\nabla \cdot \boldsymbol{u} = \text{div}(\boldsymbol{u}) = 0$, so the field is divergence-free by design. Richter-Powell et al. (2022) consider all input and output variables in this process, calculating the divergence with respect to the entire input vector $(t, \boldsymbol{x})$. In our case, we generalized their approach to produce divergence-free fields with respect to any subset of the input and output variables, as long as they are of the same size. See Appendix B.1 for more details and examples.

## 2.3 A Practical Example: Navier-Stokes Equations

Having introduced all the components of CP, we now go through a practical example of its application. We consider the 2D Navier-Stokes equations:

$$\begin{cases} \frac{\partial \boldsymbol{u}}{\partial t} + [\mathrm{D}_{\boldsymbol{x}} \boldsymbol{u}] \cdot \boldsymbol{u} - \Delta \boldsymbol{u} + \nabla p = 0, & \text{momentum eqn.} \\ \nabla \cdot \boldsymbol{u} = 0. & \text{incompressibility eqn.} \end{cases} \tag{6}$$

To solve this system with CP, we consider two modules. The first one is a PINN and learns the momentum equation in 6, predicting $(\hat{\boldsymbol{u}}_1, \hat{p}_1)$. Its specific loss components in equation 2 are

$$\begin{aligned} \mathcal{L}_{\text{BC 1}} &= \|\hat{\boldsymbol{u}}_1 - \boldsymbol{u}\|_2^2 + \|\hat{p}_1 - p\|_2^2 \\ \mathcal{L}_{\text{IC 1}} &= \|\hat{\boldsymbol{u}}_1(0, \boldsymbol{x}) - \boldsymbol{u}(0, \boldsymbol{x})\|_2^2 + \|p_1(0, \boldsymbol{x}) - p(0, \boldsymbol{x})\|_2^2 \\ \mathcal{L}_{\text{PDE 1}} &= \left\| \frac{\partial \hat{\boldsymbol{u}}_1}{\partial t} + [\mathrm{D}_{\boldsymbol{x}} \hat{\boldsymbol{u}}_1] \cdot \hat{\boldsymbol{u}}_1 - \Delta \hat{\boldsymbol{u}}_1 + \nabla \hat{p}_1 \right\|_2^2. \end{aligned} \tag{7}$$

The second module, which is either a PINN or an NCL, is related to the incompressibility equation in 6, and predicts only $\boldsymbol{u}$. If the module is a PINN, the loss components are

$$\mathcal{L}_{\text{BC 2}} = \|\hat{\boldsymbol{u}}_2 - \boldsymbol{u}\|_2^2, \qquad \mathcal{L}_{\text{IC 2}} = \|\hat{\boldsymbol{u}}_2(0, \boldsymbol{x}) - \boldsymbol{u}(0, \boldsymbol{x})\|_2^2, \qquad \mathcal{L}_{\text{PDE 2}} = \|\nabla \cdot \hat{\boldsymbol{u}}_2\|_2^2, \tag{8}$$

while $\mathcal{L}_{\text{PDE 2}}$ is omitted if this module is instead an NCL, as it is satisfied by design. To define the alignment loss, we identify the common variables of the two modules, that is $\boldsymbol{u}$. Hence, this loss is defined as $\mathcal{L}_{\text{align}} = \|\hat{\boldsymbol{u}}_1 - \hat{\boldsymbol{u}}_2\|_2^2$. During evaluation, we need to predict all variables that define the problem: $\boldsymbol{u}$ and $p$. Since the first module predicts both, it is the only module required to evaluate CP.

## 2.4 Theoretical Considerations

Theoretical results on the convergence of physical models (like PINN and NCL) to the true solution remain an open problem. Some insights exist only on a limited class of single PDEs (Yeonjong Shin et al., 2020). Systems of PDEs represent a harder problem, as PINNs often fail to learn the underlying solution. This is especially true for the Navier-Stokes equation, where the turbulent behavior of fluids represents an issue for these models (Sun et al., 2020; Xiang et al., 2022).

Since the Sobolev distance is a widely used tool for the convergence, existence, or uniqueness of solutions to PDEs (Evans, 2022), we expect our derivative-based alignment (SOB and DERL) to outperform the OUTL alignment.

Intuitively, if the SOB/DERL alignment loss were always zero, the aligned modules would predict the same solution. This would make our modular CP equivalent to a single learning model that has access to the overall system. However, in this case CP would enjoy a simpler optimization problem,

Table 1: Summary of the tasks we consider in the experiments. The last column indicates the different CP configurations, in terms of the modules, tested in the specific experiment.

| Experiment | Equation | Description | CP Modules |
|---|---|---|---|
| **Navier-Stokes Taylor-Green vortex** | (TG.M) $\dfrac{\partial \boldsymbol{u}}{\partial t} + [\mathrm{D}_x \boldsymbol{u}] \cdot \boldsymbol{u} - \Delta \boldsymbol{u} + \nabla p = 0$ 
 (TG.I) $\nabla \cdot \boldsymbol{u} = 0$ | Time dependent 2 PDEs | 2 PINNs (2xPINN) PINN+NCL |
| **Navier-Stokes Kovasznay flow** | (KF.M) $[\mathrm{D}_x \boldsymbol{u}] \cdot \boldsymbol{u} - \Delta \boldsymbol{u} + \nabla p = 0$ 
 (KF.I) $\nabla \cdot \boldsymbol{u} = 0$ | Time independent 2 PDEs | 2 PINNs (2xPINN) PINN+NCL |
| **Acoustics equations** | (A.P) $\dfrac{\partial p}{\partial t} + K \left( \dfrac{\partial u}{\partial x} + \dfrac{\partial v}{\partial y} \right) = 0,$ 
 (A.Vx) $\dfrac{\partial u}{\partial t} + \dfrac{1}{\rho} \dfrac{\partial p}{\partial x} = 0,$   (A.Vy) $\dfrac{\partial v}{\partial t} + \dfrac{1}{\rho} \dfrac{\partial p}{\partial y} = 0$ | Time dependent 3 PDEs | 3 PINNs (3xPINN) 3 NCLs (3xNCL) |
| **Navier Stokes Euler (NS-Euler) Gas equations** | (EG.C) $\dfrac{\partial \rho}{\partial t} + \nabla \cdot (\rho \boldsymbol{u}) = 0$   (EG.I) $\nabla \cdot \boldsymbol{u} = 0$ 
 (EG.M) $\dfrac{\partial \boldsymbol{u}}{\partial t} + [\mathrm{D}_x \mathbf{u}] \boldsymbol{u} + \dfrac{\nabla p}{\rho} = 0$ | Time dependent 3 PDEs | 2 PINNs (2xPINN) PINN+NCL 2 NCLs (2xNCL) 3 PINNs (3xPINN) |
| **MagnetoHydroDynamics (MHD)** | (MHD.C) $\dfrac{\partial \rho}{\partial t} + \nabla \cdot (\rho \boldsymbol{u}) = 0$   (MHD.G) $\nabla \cdot \boldsymbol{B} = 0$ 
 (MHD.M) $\rho \left( \dfrac{\partial \boldsymbol{u}}{\partial t} + [\mathrm{D}_x \mathbf{u}] \boldsymbol{u} \right) = (\nabla \times \boldsymbol{B}) \times \boldsymbol{B} - \nabla p$ 
 (MHD.I) $\dfrac{\partial \boldsymbol{B}}{\partial t} = \nabla \times (\boldsymbol{u} \times \boldsymbol{B})$   (MHD.S) $\dfrac{\mathrm{d}}{\mathrm{d}t} \left( \dfrac{p}{\rho^\gamma} \right) = 0$ | Time dependent five PDEs | 2 PINNs (2xPINN) 2xPINN+NCL 3 PINNs (3xPINN) 4 PINNs (4xPINN) |

as each module would only focus on one equation. Furthermore, with the simplified alignment loss, each module optimizes the IC, BC, one PDE loss, and one term of the alignment loss on average, for a total of 4. Instead, a model optimized on the full system of $n$ PDEs involves $n + 2$ loss terms. Hence, from 3 PDEs onwards, CP has this further advantage. In Section 3.4, we analyze how the different loss terms backpropagate through the different layers of the modules, showing that those of CP have much more similar scales than those of PINNs, improving the quality of training steps. In Appendix G, we provide further intuitions on the differences between alignment and PDE residual losses, as well as a more explicit example on the transfer of physical constraints between modules.

## 3 EXPERIMENTS: CASE STUDIES

We empirically validate the effectiveness of CP on several physical systems of PDEs. We work with systems of 2, 3 and 5 PDEs. Table 1 provides a summary of the tasks and the related CP configuration. In this Section, we focus on the Navier-Stokes and Acoustic equations as the first case studies. At the end, we also provide an empirical explanation of why CP outperforms state-of-the-art approaches by analyzing the gradients backpropagated through CP modules. Section 4 evaluates CP on two challenging real-world systems: NS-Euler Gas and MagnetoHydroDynamics (MHD) equations[1].

We compare CP with four main approaches: PINN (Raissi et al., 2019), PINN with gradient based reweighting (PINN+Grad, Wang et al. (2023)), PINN with RAR point resampling (PINN+RAR, Wu et al. (2023)), and NCL (Richter-Powell et al., 2022). Further details are available in Appendix B. These models are implemented as a single model that accesses the information about all PDEs. This is done by replacing the last term in Equation 5 with $\sum_{i=1}^{N} \|\mathcal{F}_{\text{PDE } i}[\hat{\boldsymbol{u}}]\|_2^2$. We remark that NCL satisfies divergence-free equations by design, without requiring any loss term. We remark that NCL is a generalization of the original approach of Raissi et al. (2019) for divergence-free fields. For completeness, we also test CP with gradient-based reweighting.

Following Wang et al. (2023), we train all models for $100,000$ steps with batches of $1000$ points randomly sampled in the domain. We use the Adam optimizer (Kingma, 2014) with an initial learning rate of $0.001$, decaying every $2000$ steps by a factor of $0.9$. Model architectures are in Appendix C.3.

To measure the accuracy of the predictions, we evaluate all approaches with two metrics: (i) $\|\boldsymbol{u} - \hat{\boldsymbol{u}}\|_2$ ($L^2$-err), which measures the $L^2$ distance between the true solution $\boldsymbol{u}$ and the estimate $\hat{\boldsymbol{u}}$. (ii) $\max_{(t,\boldsymbol{x})} |\hat{\boldsymbol{u}} - \boldsymbol{u}|$ (max-err), which measures the maximum absolute error in the domain.

---

[1]The code to reproduce the experiments is available at https://github.com/AlexThirty/ComPhy

Table 2: Results for the Taylor-Green vortex, Kovasznay flow, and Acoustics experiments. $L^2$ errors of the model predictions with respect to the ground truth, and the maximum errors in the domain.

| Model | Alignment | Taylor-Green Vortex $L^2$-err. $\times 10^{-5}$ | max err. $\times 10^{-3}$ | Kovasznay Flow $L^2$-err. $\times 10^{-6}$ | max err. $\times 10^{-3}$ | Model | Alignment | Acoustics $L^2$-err. $\times 10^{-5}$ | max_err $\times 10^{-1}$ |
|---|---|---|---|---|---|---|---|---|---|
| **PINN** | | 3.677 | 9.374 | 10.83 | 5.203 | **PINN** | — | 8.016 | 2.277 |
| **PINN+RAR** | | 5.235 | 6.160 | 11.48 | 8.062 | **PINN+RAR** | — | 120.6 | 20.05 |
| **PINN+Grad** | | 4.245 | 9.719 | 7.471 | 4.264 | **PINN+Grad** | — | 10.05 | 3.391 |
| **NCL** | | 2.830 | **5.947** | **3.014** | 1.398 | **NCL** | — | 5.243 | 1.552 |
| **CP-2xPINN** | OUTL | 6.028 | 9.945 | 20.47 | 4.972 | **CP-3xPINN** | OUTL | 11.43 | 8.437 |
| **CP-2xPINN** | SOB | 5.222 | 8.027 | 8.641 | 6.195 | **CP-3xPINN** | SOB | 7.611 | 2.148 |
| — | **+Grad** | 4.687 | 6.250 | 7.695 | 4.207 | — | **+Grad** | 5.726 | 1.685 |
| **CP-2xPINN** | DERL | 3.619 | 6.378 | 7.221 | 4.996 | **CP-3xPINN** | DERL | 7.449 | 2.171 |
| — | **+Grad** | 4.311 | 6.250 | 5.370 | 4.107 | — | **+Grad** | 5.718 | 1.737 |
| **CP-PINN+NCL** | OUTL | 6.024 | 11.55 | 13.30 | 4.515 | **CP-3xNCL** | OUTL | 6.008 | 4.549 |
| **CP-PINN+NCL** | SOB | 4.764 | 9.215 | 6.867 | 3.619 | **CP-3xNCL** | SOB | 5.172 | 1.521 |
| — | **+Grad** | 4.386 | 8.418 | 3.663 | **1.277** | — | **+Grad** | 2.727 | **1.090** |
| **CP-PINN+NCL** | DERL | 3.247 | 7.996 | 5.422 | 3.468 | **CP-3xNCL** | DERL | 5.165 | 1.547 |
| — | **+Grad** | **2.795** | 6.468 | 4.325 | 1.513 | — | **+Grad** | **2.718** | 1.121 |

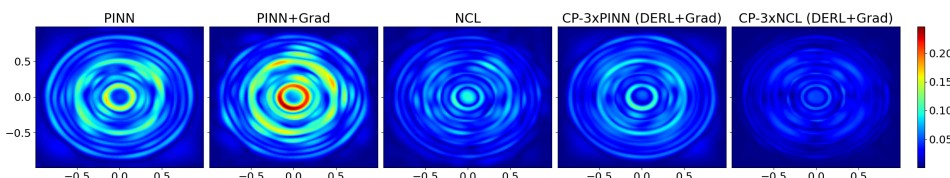

Figure 2: Acoustics equation: errors for PINN, PINN+Grad, NCL, and the best CP models (one for architecture) at $t = 0.16$. The lowest errors are in blue.

## 3.1 TAYLOR-GREEN VORTEX

Our first experiment involves the 2D time-dependent incompressible Navier-Stokes equations, namely the momentum (TG.M) and incompressibility (TG.I) Equations in Table 1. We work in the domain $[0, \pi] \times [0, \pi]$ with time $t \in [0, 10]$ and a viscosity coefficient of $\nu = 0.1$. The analytic solution of the Taylor-Green vortex (Chorin, 1968), used to calculate the reference solution, is given in Appendix D.1. The inference module for CP is the one learning equation (TG.M), as it predicts all the variables.

Table 2 shows the numerical results for this experiment, while Appendix D.1 shows the prediction error for different values of $t$. We see that CP models aligned with DERL outperform all variations of PINNs. CP-PINN+NCL with DERL alignment and Grad reweighting performs best in the $L^2$ metric, even when compared to NCL. This is remarkable, since NCL has the built-in advantage of bypassing the divergence-free equation, while CP has to transfer it across modules. Additional results are available in Appendix D.1.

## 3.2 KOVASZNAY FLOW

We consider the 2D time-independent Navier-Stokes equations made of the momentum (KF.M) and incompressibility (KF.I) equations in Table 1. We work in the domain $[-1, 1] \times [-0.5, 1.5]$ with a viscosity coefficient of $\nu = \frac{1}{50}$. The analytic solution of the Kovasznay flow (Drazin & Riley, 2009), which represents the flow behind a two-dimensional grid, is given in Appendix D.2. The evaluation model for CP is always the one for equation (KF.M), as it predicts all the variables. Numerical results are provided in Table 2. The complete set of results and figures is available in Appendix D.2. None of the PINN variants matches the performance of the CP models aligned with either DERL or SOB. Interestingly, OUTL alignment delivers worse results, supporting our hypothesis that derivative-based alignments are more effective. Due to its implicit advantage of modeling divergence-free fields, NCL shows a competitive performance. However, CP with gradient-based reweighting obtains the best performance possible, between 2 and 3 times better than PINN variants. We conclude that within these Navier-Stokes case studies, ComPhy alleviates PINN optimisation issues and performs comparably to, if not better than, NCL. However, CP is far more general than NCL, and can be

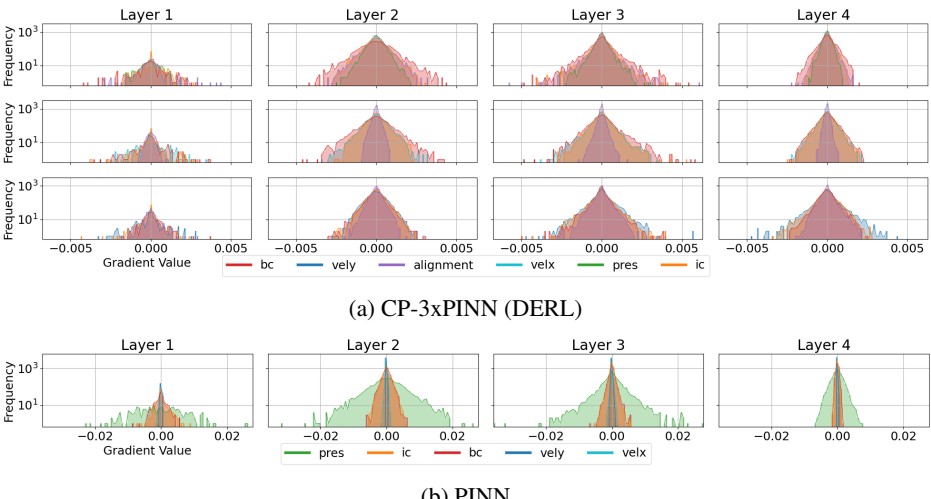

Figure 3: Acoustics experiment gradient histograms. Each plot contains the histograms for the distribution of the gradients propagated at each layer of the CP-3xPINN and PINN models at the beginning of training, similarly to Wang et al. (2021). Different colors are for different losses.

directly applied to systems without divergence-free components. As OUTL consistently showed worse results, for the remaining experiments, we only work with derivative-based alignment.

### 3.3 ACOUSTIC EQUATIONS

We now consider a set of three equations, namely equations (A.P), (A.Vx), and (A.Vy) in Table 1, solved using CP model with three aligned modules. As we can see, all three equations are in the divergence-free form (see Appendix B.1), respectively for $t, x, y$ for the first equation, $t, x$ and $t, y$ for the second and third. Hence, we can employ an CP model with three NCL modules. Since equation (A.P) contains all relevant variables, the first module is the one used for evaluation.
We work in the spacetime domain of $(t, x, y) \in [0, 0.24] \times [-1, 1]^2$, with density $\rho \equiv 1$ and bulk coefficient $K \equiv 1$. Appendix D.3 provides the initial conditions, while boundary conditions are null everywhere. The reference solution is calculated using the Clawpack software package (Clawpack Development Team, 2024; Mandli et al., 2016). Further details can be found in Appendix D.3.

We report the numerical results in Table 2, while Figure 2 shows the prediction error in the domain at $t = 0.16$. For additional Figures, see Appendix D.3. Even when using 3 different modules, CP is clearly the best approach for both our metrics. In particular, CP with 3 NCL modules performs best. None of the PINN variants fall close to our best model, not even when adding resampling or adaptive weights. CP, instead, benefits from gradient-based reweighting, showing a 2x/3x improvement with respect to PINNs and NCL. This experiment suggests that CP and derivative-based alignment scales well to a larger number of modules. Section 4 scales CP to even larger systems.

### 3.4 GRADIENT ANALYSIS

To get a better understanding of why CP outperforms PINNs, we conduct an empirical analysis on the distribution of the backpropagated gradients at different layers of the CP modules. Given $\boldsymbol{w}_j$ the set of weights in layer $j$, for each loss term $\mathcal{L}_{\text{BC}}, \mathcal{L}_{\text{IC}}, \mathcal{L}_{\text{PDE}\,i}$ of equation 2, we collect the gradients $\frac{\partial \mathcal{L}}{\partial \boldsymbol{w}_j}$ and we plot the corresponding histogram grouped by layer (Figure 3). A similar analysis for PINNs is discussed in Wang et al. (2021). The authors show that when PINNs are optimized correctly, the gradients of the different losses tend to be evenly distributed within each layer. However, if the distributions differ significantly, it implies that one loss function is dominating, which negatively affects the model's performance.

We replicate this study for CP on the Acoustics experiment of Section 3.3, which involves 3 equations and 3 modules. Figures 3a and 3b show the distributions of propagated gradients for CP 3xPINN

Table 3: Numerical results for the Euler gas and MHD equation experiments. We report $L^2$ errors of the model predictions with respect to the ground truth, and the maximum errors in the domain.

(a) Euler gas equation experiment.

| Model | Alignment | $L^2$-err. $\times 10^{-3}$ | max_err $\times 10^0$ |
|---|---|---|---|
| **PINN** | — | 1.712 | 5.364 |
| **PINN+Grad** | — | 3.319 | 7.127 |
| **NCL** | — | 1.690 | 4.912 |
| **CP-2xPINN** | SOB | **1.296** | 3.082 |
| **CP-2xPINN** | DERL | 1.401 | 3.149 |
| **CP-PINN+NCL** | SOB | 1.487 | 2.721 |
| **CP-PINN+NCL** | DERL | 1.468 | **2.673** |
| **CP-2xNCL** | SOB | 2.029 | 4.311 |
| **CP-2xNCL** | DERL | 1.396 | 4.183 |
| **CP-3xPINN** | SOB | 1.382 | 3.021 |
| **CP-3xPINN** | DERL | 1.462 | 2.899 |

(b) MagnetoHydroDynamics experiment.

| Model | Alignment | $L^2$-err. $\times 10^{-4}$ | max_err $\times 10^{-1}$ |
|---|---|---|---|
| **PINN** | — | 1.967 | 12.74 |
| **PINN+Grad** | — | 1.657 | 9.164 |
| **NCL** | — | 1.975 | 12.59 |
| **CP-2xPINN** | SOB | 1.599 | 9.324 |
| **CP-2xPINN** | DERL | 1.608 | 9.280 |
| **CP-3xPINN** | SOB | **1.535** | **8.528** |
| **CP-3xPINN** | DERL | **1.535** | **8.548** |
| **CP-2xPINN+NCL** | SOB | 1.601 | 8.913 |
| **CP-2xPINN+NCL** | DERL | 1.607 | 8.934 |
| **CP-4xPINN** | SOB | 1.560 | 8.764 |
| **CP-4xPINN** | DERL | 1.567 | 8.771 |

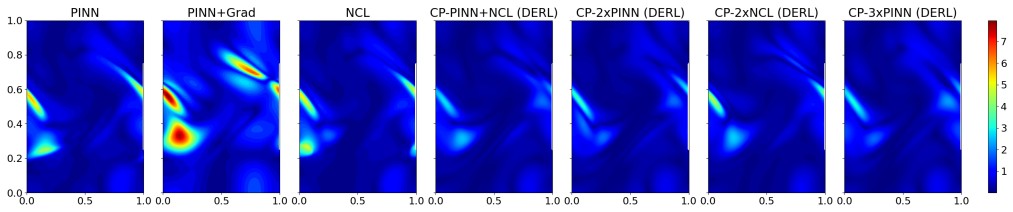

Figure 4: NS-Euler gas equations experiment: prediction error in the domain at $t = 0.3$ for the PINN, NCL, and the CP model (best for each architecture).

and PINN, respectively. The plots clearly highlight that combining three modules makes the gradient distribution more aligned across layers. As we have seen in Section 3.3, this also entails a better performance. We provide the same analysis on the Kovasznay flow experiment in Appendix D.2. Both studies support the fact that the modular approach of ComPhy results in an easier optimization problem when compared to a single PINN trained on the entire system of PDEs.

## 4 EXPERIMENTS: REAL-WORLD SYSTEMS

### 4.1 NS-EULER EQUATIONS

We now challenge CP with two real-world systems: the NS-Euler and the MagnetoHydroDynamics equations. The NS-Euler equations for an incompressible gas are given by the conservation (EG.C), incompressibility (EG.I), and momentum (EG.M) equations in Table 1, where $\rho$ is the density, $\boldsymbol{u}$ the velocity and $p$ the pressure. Our setup is the same as the one in Richter-Powell et al. (2022): we consider the 2D torus and work with the spacetime domain $(t, x, y) \in \left[0, \frac{1}{3}\right] \times [0, 1]^2$. Appendix D.4 provides the initial conditions and further details. The reference solution is available and collected from Richter-Powell et al. (2022), and was calculated via the Finite Element Method (Anderson et al., 2020)All models are trained for $600,000$ steps with batches of $1000$ random points at each step. The optimal hyperparameters and architectures are taken from Richter-Powell et al. (2022). On the NS-Euler equations we test 4 CP configurations: **CP-2xPINN**, **CP-PINN+NCL**, **CP-2xNCL**, and **CP-3xPINN**. For the first three, the first module (either a PINN or NCL) learns both equations (EG.C) and (EG.M), while the second learns only (EG.I). Since all relevant variables are in equation (EG.M), only the first module is used for evaluation.

Numerical results are provided in Table 3a, while Figure 4 shows the prediction error at $t = 0.3$. Appendix D.4 shows the model errors for different values of $t \in [0, \frac{1}{3}]$. While Richter-Powell et al. (2022) claim that the PINN fails to converge to the true solution, we did not encounter the same issue. Our PINN can solve the system to some extent. CP achieves state-of-the-art results when using both

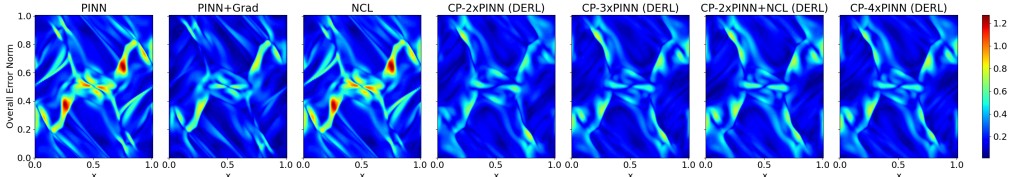

Figure 5: MHD experiment: prediction error in the domain at $t = 0.5$ for the PINN, NCL, and the CP model (best for each architecture).

DERL and SOB, with the latter struggling only when 2 NCL modules are employed. This highlights the role played by the outputs' derivatives in the alignment: a loss computed only from outputs' derivatives (DERL) is much more effective than a loss computed also from the output (SOB).
We conclude that CP can learn a challenging real-world system of PDEs effectively, using different combinations of NCL and PINN models.

## 4.2 MAGNETOHYDRODYNAMICS

Finally, we scale our model to a system of $5$ equations, the MHD equations in Table 1, which represent the motion of plasma with a combination of the compressible Navier-Stokes equations for fluids with the Maxwell equations of Magnetism (Gruber & Rappaz, 1985). These are composed of the continuity (MHD.C), momentum (MHD.M), state (MHD.S), induction (MHD.I), and Gauss (MHD.G) equations. We consider a similar setting to Gopakumar et al. (2025), with domain $[0, 1]^2$ and time in $t \in [0, 0.5]$. The reference solution is calculated with the finite volume method (Ferziger & Peric, 2001). Since we are interested in pure scaling, we consider combinations of $2, 3$ and $4$ PINNs for our CP models, as well as 2xPINN+NCL. We compare to a PINN, PINN+Grad, and an NCL with equation (MHD.G) satisfied by design. Architectures are available in Appendix C.3. Models are trained for $100, 000$ steps with the Adam optimizer, and batches of $1024$ randomly sampled points.

We report the results in Table 3b, with Figure 5 showing the prediction error at $t = 0.5$. Even with $5$ equations, CP outperforms PINN and NCL by a large margin with any of the tested combinations of modules, including that with $4$ modules. We report additional results and Figures in Appendix D.5.

## 4.3 DISCUSSION AND FURTHER EXPERIMENTAL ANALYSIS

**On the choice of the CP modules.** Apart from the MHD equations, where we tested the scaling capabilities of CP to a higher number of PDEs, we experimented with every possible combination of modules, keeping in mind that the NCL module requires a divergence-free equation to be applied. Our results support the fact that, when possible, a combination of modules that includes NCL is always beneficial. Similarly, since most physical systems present 2 or 3 equations, the most common and effective choice is to use 2 or 3 modules, respectively.

**On the strength of the alignment process.** In Section 2.4 and Appendix G, we provided theoretical understandings on why the alignment mechanism allows for effective transfer of physical constraints between modules. When the alignment loss converges to zero, the corresponding modules behave as the same function in the Sobolev space $W^{1,2}$ (Trenta et al., 2026). To empirically validate these claims, we plot the curves of the alignment loss, the prediction loss, and the PDE residuals that are not actively trained on the inference module. We also empirically measure the distance and relative discrepancy between the predictions of different modules. The results, available in Appendix E, confirm our theoretical claims and the strength of the alignment mechanism: modules predict very similar functions, while prediction and PDE losses are correlated with the alignment loss.

**Ablations.** To test the robustness of ComPhy in different setups and conditions, we performed ablation studies available in Appendix F. (i) We tried different values for the alignment coefficient $\lambda_{align}$, showing that the naive choice of $\lambda_{align} = 1$ works well on average, with possible improvements with highly specialised tuning of this parameter. We remark that the Gradient-based reweighting scheme used in Section 3 alleviates the burden of hyperparameter selection. (ii) We incremented the number of units in the MLPs of the PINN and NCL baselines to match the parameter budget of

CP during training. This leads to fewer parameters used by CP during inference, as it employs only one module at that state. Even so, CP performs better, showing the importance of modularization for these applications. (iii) We run the same experiment multiple times with different seeds to test the robustness to initializations and stochastic gradient descent. The results remain strong, as the CP models have the best mean results by far, and a lower standard deviation.

## 5 RELATED WORKS

PINNs (Raissi et al., 2019) are widely used to solve physical problems such as fluid dynamics (Sun et al., 2020). However, PINNs suffer from optimization problems, such as imbalances in the gradients of the different loss terms (Wang et al., 2022) or the ill-conditioning and complex loss landscape of PDE residual terms (Krishnapriyan et al., 2021). PINNs can also fail to reach a minimum of the loss (Sun et al., 2020). Recent approaches try to alleviate these problems with resampling strategies that explore regions with higher PDE residuals while retaining enough points in well-optimized regions (Daw et al., 2023), or by using a variational formulation of the PDE residual term in small regions around collocation points (Wu et al., 2024). Other works adopt strategies to align the gradients at different gradient descent steps using second-order corrections (Wang et al., 2025), which can be computationally demanding, or to align the autoregressive updates to the underlying true dynamics (Zhu et al., 2025) for data-driven models. Instead, we work in an unsupervised setting, and our alignment imposes on the different modules to share information on their respective physical constraints. Even though many other solutions, such as resampling or reweighting schemes, have been proposed (Sun et al., 2020; Wang et al., 2021; Xiang et al., 2022; McClenny & Braga-Neto, 2023; Zhao, 2021; Lau et al., 2024), none of them is tailored to tackle systems of PDEs.

Ad-hoc architectures solve the optimization issues by satisfying physical properties by design. NCL (Richter-Powell et al., 2022) ensures that the output of the model is divergence-free, while Torres et al. (2024) proposes a divergence-free normalizing flow. However, such solutions only work for a given physical property and still require enforcing additional ones with alternative approaches, like PINN. Finally, our CP is the first attempt at solving systems of PDEs with a modular, end-to-end approach where modules are aligned to transfer physical constraints with each other.

Neural Operators (Kovachki et al., 2023) are deep models that learn mappings between functional spaces. DeepONets (Lu et al., 2021) try to learn representations of the input function and combine them on the output function inference points. Graph Neural Operators (Li et al., 2020), instead, transform functions via local convolutions with parametrized kernels, which are learned during training. Finally, Fourier Neural Operators (Li et al., 2021) use Fourier transforms to simplify and speed up the calculation of such convolutions while acting on the function globally. While these methods are powerful for supervised learning tasks and generalization to different initial conditions or parameters, they are more computationally demanding and serve a different task, as they learn mappings between functions. Furthermore, they need simulated data, and they are *not* suitable for the unsupervised setting of PINNs.

## 6 CONCLUSION

We introduced CP, a modular approach for solving systems of PDEs where the final predicted solution is the result of the collective effort of the modules. The modules communicate with each other during training via alignment, a process that encourages the predictions of the same variable made by different modules to be similar. In particular, we consider a derivative-based alignment, and we show that it provides state-of-the-art results on several systems of PDEs, including challenging real-world tasks. We further analyzed why CP optimizes the physics-informed losses better than PINNs, finding that their propagated gradients have more similar scales. While NCL is competitive in the case studies, the advantage of having one equation less to be optimized fades away in real-world systems of three or more equations, where CP scales much better.

Future works could consider different optimization protocols, such as alternating module optimization steps with alignment steps, as well as aligning higher-order derivatives to capture richer dynamics of the physical system. CP represents an initial step toward developing modular architectures for scalable learning of physical systems. Progress in this direction could lead to dynamic models capable of adapting to new initial conditions, as well as incorporating or removing modules over time to accommodate changing environments.

ACKNOWLEDGMENTS

This work has been supported by EU-EIC EMERGE (Grant No. 101070918).

ETHICS STATEMENT

The research conducted in this work completely adheres to the ICLR Code of Ethics. This work does not involve human subjects, nor does it provide harmful insights, discrimination/bias concerns, or any other potential ethical concerns.

REPRODUCIBILITY STATEMENT

We provide all the necessary information to reproduce the results in the paper in the main text and in the Appendix. The methodology is described in Section 2, while experimental settings, along with training steps, optimizers, and model architectures, are described in Section 3 and Section 4. Further implementation details are available in Appendix C. We additionally provide all the code necessary to reproduce the experiments, from data generation to model definition and training, as well as evaluation, as Supplementary Material. The code is also available at https://github.com/AlexThirty/ComPhy

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

## A LLM USAGE

Large Language Models (LLMs) were used to polish and check the grammar of the paper, with very limited usage to rewrite some phrases for improved readability. No LLM was used to write entire parts of the text or for any other non-specified purposes. Every output of the LLMs was carefully checked to ensure its correctness and validity, adhering to the LLM policy and Code of Ethics.

## B MODEL DESCRIPTION

This Section provides further information on CP models and their components, as well as the employed baselines.

### B.1 PARTICULAR NCL APPLICATIONS

While PINNs have a straightforward implementation, NCL models require performing a final transformation on the network's output. In particular, the MLP parametrizes an antisymmetric matrix $\mathbf{A}$, which depends on *all* the input coordinates $(t, x, y)$. As described in Section 2, a row-wise divergence is performed to obtain a vector that has zero divergence with respect to *all* the inputs. To obtain a vector field that is divergence-free with respect to *only a subset* of the input coordinates, it is sufficient to consider a smaller matrix $\mathbf{A}$ and apply the above procedure only for the interested coordinates. We provide some examples here.

In the case of the 2D time-dependent Navier-Stokes equation, the incompressibility equation is given by

$$\nabla \cdot \boldsymbol{u} = \text{div}_{(x,y)}(\boldsymbol{u}) = \frac{\partial u_x}{\partial x} + \frac{\partial u_y}{\partial y} = 0, \tag{9}$$

where the field $\boldsymbol{u}$ depends on the three inputs $(t, x, y)$. Here, the MLP parametrizes $\mathbf{A}$ as a $2 \times 2$ antisymmetric matrix depending on $(t, x, y)$. The row-wise divergence is taken only with respect to $(x, y)$, so that the final output is divergence-free in just those variables while depending freely on $t$. In the case of the NS-Euler equations, the mass conservation for the density $\rho$ is given by

$$\frac{\partial \rho}{\partial t} + \nabla \cdot (\rho \boldsymbol{u}) = \text{div}_{(t,x,y)}(\rho, \rho \boldsymbol{u}) = 0. \tag{10}$$

To achieve this, the MLP parametrizes $\mathbf{A}$ as a $3 \times 3$ antisymmetric matrix and the row-wise divergence is taken with respect to all three variables $(t, x, y)$. While a divergence-free equation is guaranteed, other constraints can be added via losses similar to those of the PINN model. Finally, we consider the non-trivial case of the NCL model for the $y$-velocity equation in the Acoustics equations experiment in Section 3.3, which reads

$$\frac{\partial v}{\partial t} + \frac{\partial p}{\partial y} = 0. \tag{11}$$

To achieve this, it is sufficient for the MLP to output a $2 \times 2$ antisymmetric matrix (which is parametrized by only one output). Then, $v$ and $p$ can be calculated as

$$\begin{aligned} v &= \text{div}_{(t,y)}\left(\mathbf{A}_{0\cdot}\right) = \frac{\partial A_{00}}{\partial t} + \frac{\partial A_{01}}{\partial y}, \\ p &= \text{div}_{(t,y)}\left(\mathbf{A}_{1\cdot}\right) = \frac{\partial A_{10}}{\partial t} + \frac{\partial A_{11}}{\partial y}. \end{aligned} \tag{12}$$

This way, the vector field $(v, p)$ predicted by the NCL depends on all three variables $(t, x, y)$, but is divergence-free with respect to only $(t, y)$. While this example is still simple to write and involves 2 variables, the above process can be generalized to any number and type of coordinates and for every possible subset of them.

### B.2 SOBOLEV NORMS AND ALIGNMENT MODULES

In the mathematics literature, Sobolev spaces (Maz'ya, 2011) are the common choice to study and prove the existence and uniqueness of solutions to PDE problems (Evans, 2022). From a theoretical point of view, it is not enough to measure distances in the $L^2$, especially for convergence results. In

addition, not all PDE problems admit smooth solutions, and a notion of weak derivative is needed; hence, the necessity to consider a more restrictive space, that is, the Sobolev space $W^{p,m}(\Omega)$ of functions $u(\boldsymbol{x})$ with bounded norm

$$\|u(\boldsymbol{x})\|_{W^{p,m}(\Omega)} = \|u(\boldsymbol{x})\|_{L^2(\Omega)} + \sum_{\ell=1}^{m} \|\mathrm{D}^\ell u(\boldsymbol{x})\|_{L^2(\Omega)}, \tag{13}$$

where $\mathrm{D}^\ell u(\boldsymbol{x})$ is the $l$-th order differential of $u$. Weak derivatives of a function $u$ are defined such that they resemble integration by parts for every possible test function $v \in C_c^\infty(\Omega)$ with compact support in $\Omega$, that is

$$\int_\Omega \frac{\partial^{\alpha_1+\alpha_2+\alpha_h+\cdots} u}{\partial x_1^{\alpha_1} \, \partial x_2^{\alpha_2} \, \partial \dots \cdots \, \partial x_h^{\alpha_h}} \cdot v \; dx^n = (-1)^{\alpha_1+\dots+\alpha_h} \int_\Omega u \cdot \frac{\partial^{\alpha_1+\alpha_2+\alpha_h+\cdots} v}{\partial x_1^{\alpha_1} \, \partial x_2^{\alpha_2} \, \partial \dots \cdots \, \partial x_h^{\alpha_h}} dx^n \tag{14}$$

Furthermore, while the derivatives and the boundary and initial conditions completely determine a function $u$, and are therefore sufficient to learn $u$ (Trenta et al., 2026), information on $u$ is not sufficient to determine its derivatives, which are fundamental for the dynamics and evolution of a system. In practice, using the Sobolev norm to train a neural network has been shown to improve the performance, especially with low data availability (Czarnecki et al., 2017). In the context of physical systems, the derivatives are fundamental to ensure the physical system is learned and also to ensure the correct transfer of physical information between models (Trenta et al., 2026). For these reasons, the alignment module must contain the derivatives of the networks to ensure that the dynamics of the system, predicted by the two networks, are coherent. This explains why the alignment module with only $u$ (OUTL) performs so poorly compared to DERL and SOB.

## B.3 PDE RESIDUALS AND PINNS

In this Section, we provide more details on PDEs and their residuals. A PDE of order $d$ is, in general, an expression that involves one or more partial derivatives, up to order $d$ of a function $u$:

$$\mathcal{F}[u] = F\left(u, \mathrm{D}u, \mathrm{D}^2 u, \dots, \mathrm{D}^d u\right) = 0. \tag{15}$$

In the linear case, for example, it can be written as

$$\mathcal{F}[u](\boldsymbol{x}) = f(\boldsymbol{x}) + a_0(\boldsymbol{x})u(\boldsymbol{x}) + \sum_{i=1}^{n} a_i(\boldsymbol{x}) \frac{\partial u}{\partial x_i}(\boldsymbol{x}) + \sum_{i=1}^{n} \sum_{j=1}^{n} a_{ij}(\boldsymbol{x}) \frac{\partial^2 u}{\partial x_i \, \partial x_j}(\boldsymbol{x}) + \dots \tag{16}$$

PINNs implement the PDE itself as a soft target in the loss. To do so, the PDE operator $\mathcal{F}$, which takes as input a function and outputs another function, which we aim to be null, is calculated on the MLP itself. Derivatives of the MLP $\hat{u}, \mathrm{D}\hat{u}, \dots$ are calculated using Automatic Differentiation (Baydin et al., 2018), and the function $F$ is calculated with them, obtaining the current PDE residual of the MLP itself

$$\mathcal{F}[\hat{u}] = F\left(\hat{u}, \mathrm{D}\hat{u}, \mathrm{D}^2 \hat{u}, \dots, \mathrm{D}^d \hat{u}\right). \tag{17}$$

The norm of this residual is calculated and used to optimize the network. Ideally, if this loss term is $0$ or close to it, the model itself should satisfy the PDE. This is also why we analyze it in Section 3.4, to see how well a model is consistent with the underlying equation. Coupled with the loss for the initial and boundary conditions, this is the main component of a Physics-Informed Neural Network (Raissi et al., 2019).

## B.4 GRADIENT-BASED REWEIGHTING

Gradient-based reweighting is a technique that automatically selects hyperparameters for the different loss terms based on the current gradient norm. We consider the approach described in Wang et al. (2023), where a model has parameters $\boldsymbol{\theta}$ and loss function

$$\mathcal{L}_{\text{tot}}(\boldsymbol{\theta}) = \lambda_{\text{IC}} \mathcal{L}_{\text{IC}}(\boldsymbol{\theta}) + \lambda_{\text{BC}} \mathcal{L}_{\text{BC}}(\boldsymbol{\theta}) + \lambda_{\text{PDE}} \mathcal{L}_{\text{PDE}}(\boldsymbol{\theta}). \tag{18}$$

Then, the optimal weights are calculated as

$$\hat{\lambda}_{\text{IC}} = \frac{\|\nabla_{\boldsymbol{\theta}} \mathcal{L}_{\text{IC}}\| + \|\nabla_{\boldsymbol{\theta}} \mathcal{L}_{\text{BC}}\| + \|\nabla_{\boldsymbol{\theta}} \mathcal{L}_{\text{PDE}}\|}{\|\nabla_{\boldsymbol{\theta}} \mathcal{L}_{\text{IC}}\|},$$

$$\hat{\lambda}_{\text{BC}} = \frac{\|\nabla_{\boldsymbol{\theta}} \mathcal{L}_{\text{IC}}\| + \|\nabla_{\boldsymbol{\theta}} \mathcal{L}_{\text{BC}}\| + \|\nabla_{\boldsymbol{\theta}} \mathcal{L}_{\text{PDE}}\|}{\|\nabla_{\boldsymbol{\theta}} \mathcal{L}_{\text{BC}}\|}, \tag{19}$$

$$\hat{\lambda}_{\text{PDE}} = \frac{\|\nabla_{\boldsymbol{\theta}} \mathcal{L}_{\text{IC}}\| + \|\nabla_{\boldsymbol{\theta}} \mathcal{L}_{\text{BC}}\| + \|\nabla_{\boldsymbol{\theta}} \mathcal{L}_{\text{PDE}}\|}{\|\nabla_{\boldsymbol{\theta}} \mathcal{L}_{\text{PDE}}\|}.$$

This formulation generalizes easily to multiple PDE terms and the alignment term.

### B.5 ADAPTIVE POINT RESAMPLING (RAR)

For our adaptive point resampling, we consider RAR (Wu et al., 2023). After a certain number of iterations, the PDE residuals of the system are calculated on a grid of points. For each PDE, the $k$ points with the highest residual are collected and added to the training data.

## C  IMPLEMENTATION DETAILS

In this section, we provide details on the practical implementation of the CP model, from the discretization of the losses to the computational times and the tuning process.

### C.1  LOSS DISCRETIZATION

Since it is not possible to calculate the true $L^2$ losses and distances for the models, we discretize them as is usually done in the literature. In general, we assume to have 4 datasets:

- The first one contains $N_D$ points in the interior of the spatio-temporal domain $[0, T] \times \Omega$ with no given supervised targets:

$$\mathcal{D}_D = \{(t_i, \boldsymbol{x}_i), t_i \in [0, T], \boldsymbol{x}_i \in \Omega, i = 1, \ldots, N_D\}. \tag{20}$$

  Points in this dataset are often called *collocation points* in literature and are used in the PDE and alignment loss. The intuition is that, if these losses are close to zero, the model satisfies the PDE (or the networks are aligned) for sufficient points in the interior of the domain. The PDE residual loss is then approximated as

$$\|\mathcal{F}[\hat{u}]\|_{L^2([0,T]\times\Omega)}^2 \approx \frac{1}{N_D} \sum_{i=1}^{N_D} \|\mathcal{F}[\hat{u}](t_i, \boldsymbol{x}_i)\|^2, \tag{21}$$

  while the alignment loss components are discretized in a similar way

$$\|\hat{u}_1 - \hat{u}_2\|_{L^2([0,T]\times\Omega)}^2 \approx \frac{1}{N_D} \sum_{i=1}^{N_D} \|\hat{u}_1(t_i, \boldsymbol{x}_i) - \hat{u}_2(t_i, \boldsymbol{x}_i)\|^2,$$

$$\|\mathrm{D}^\ell \hat{u}_1 - \mathrm{D}^\ell \hat{u}_2\|_{L^2([0,T]\times\Omega)}^2 \approx \frac{1}{N_D} \sum_{i=1}^{N_D} \|\mathrm{D}^\ell \hat{u}_1(t_i, \boldsymbol{x}_i) - \mathrm{D}^\ell \hat{u}_2(t_i, \boldsymbol{x}_i)\|^2. \tag{22}$$

- The second one contains $N_I$ points in the domain $\Omega$ at time $t = 0$ and is used to enforce the initial condition $u(0, \boldsymbol{x}) = g(\boldsymbol{x})$:

$$\mathcal{D}_I = \{((0, \boldsymbol{x}_i), g(\boldsymbol{x}_i)), \boldsymbol{x}_i \in \Omega, g(\boldsymbol{x}_i) \in \mathbb{R}^n, i = 1, \ldots, N_I\}. \tag{23}$$

  The IC loss is then discretized as

$$\|\hat{u} - g\|_{L^2(\Omega)}^2 \approx \frac{1}{N_I} \sum_{i=1}^{N_I} \|\hat{u}(0, \boldsymbol{x}_i) - g(\boldsymbol{x}_i))\|^2 \tag{24}$$

Table 4: Time in seconds required for one step of training.

| Model | Taylor-Green vortex | Kovasznay flow | Acoustics | Euler gas | MaghetoHydroDynamics |
|---|---|---|---|---|---|
| **PINN** | 0.0420 | 0.0480 | 0.0210 | 0.2097 | 0.1093 |
| **NCL** | 0.1807 | 0.1174 | 0.0618 | 0.6889 | 0.2553 |
| **2xPINN** (OUTL) | 0.1034 | 0.0840 | / | 0.2884 | / |
| **2xPINN** (SOB) | 0.1069 | 0.0947 | / | 0.3281 | 0.1657 |
| **2xPINN** (DERL) | 0.1039 | 0.0933 | / | 0.3101 | 0.1611 |
| **PI+NCL** (OUTL) | 0.1431 | 0.1562 | / | 0.5795 | / |
| **PI+NCL** (SOB) | 0.1483 | 0.1543 | / | 0.6584 | / |
| **PI+NCL** (DERL) | 0.1413 | 0.1626 | / | 0.6563 | / |
| **2xNCL** (SOB) | / | / | / | 0.6896 | / |
| **2xNCL** (DERL) | / | / | / | 0.6858 | / |
| **3xPINN** (OUTL) | / | / | 0.0854 | 0.4366 | / |
| **3xPINN** (SOB) | / | / | 0.0898 | 0.4366 | 0.2025 |
| **3xPINN** (DERL) | / | / | 0.0888 | 0.4120 | 0.2002 |
| **3xNCL** (OUTL) | / | / | 0.1222 | / | / |
| **3xNCL** (SOB) | / | / | 0.0621 | / | / |
| **3xNCL** (DERL) | / | / | 0.0699 | / | / |
| **4xPINN** (SOB) | / | / | / | / | 0.2341 |
| **4xPINN** (DERL) | / | / | / | / | 0.2207 |

- The third one contains $N_B$ points on the boundary $\partial\Omega$ and for different times in $[0, T]$. It is used to enforce the boundary conditions $\mathcal{B}[u](t, \boldsymbol{x}) = b(t, \boldsymbol{x})$:

$$\mathcal{D}_I = \{(\,(t_i, \boldsymbol{x}_i), b(t_i, \boldsymbol{x}_i)), t_i \in [0, T], \boldsymbol{x}_i \in \Omega, b(t_i\boldsymbol{x}_i) \in \mathbb{R}^n, i = 1, \dots, N_B\}. \quad (25)$$

Boundary conditions can be of many forms, such as Dirichlet BC, which impose $u(t, \boldsymbol{x}) = b(t, \boldsymbol{x})$, or Neumann BC, which impose a condition on the derivative $(\nabla u \cdot \hat{\mathbf{n}})(t, \boldsymbol{x}) = b(t, \boldsymbol{x})$, where $\hat{\mathbf{n}}$ is the normal vector at the boundary. The loss, in this example for Dirichlet conditions, is then discretized as

$$\|\mathcal{B}[\hat{u}] - b\|_{L^2([0,T] \times \partial\Omega)}^2 \approx \frac{1}{N_B} \sum_{i=1}^{N_B} \|\hat{u}(t_i, \boldsymbol{x}_i) - b(t_i, \boldsymbol{x}_i))\|^2 \quad (26)$$

- The last one, used only at evaluation, contains a set of points in the spacetime domain $(t_i, \boldsymbol{x}_i)$ and the corresponding true solution of the system $u(t_i, \boldsymbol{x}_i)$:

$$\mathcal{D}_S = \{(\,(t_i, \boldsymbol{x}_i), u(t_i, \boldsymbol{x}_i)), t_i \in [0, T], \boldsymbol{x}_i \in \Omega, u(t_i\boldsymbol{x}_i) \in \mathbb{R}^n, i = 1, \dots, N_S\}. \quad (27)$$

This dataset is used to compare the final solution of the models against a true target, never seen during training. In this case, the loss is discretized as usual with

$$\|u - \hat{u}\|_{L^2([0,T] \times \Omega)}^2 \approx \frac{1}{N_S} \sum_{i=1}^{N_S} \|u(t_i, \boldsymbol{x}_i) - \hat{u}(t_i, \boldsymbol{x}_i)\|^2 \quad (28)$$

### C.2 COMPUTATIONAL TIME AND RESOURCES

In Table 4, we provide a comparison of the computational times of the different methods on all experiments. We reported the time to perform one epoch of training for each experiment. All models were trained on a single NVIDIA H100 GPU with 80GB of memory.

### C.3 MODEL ARCHITECTURES

We report the architectures of the models employed in Table 5.

## D ADDITIONAL RESULTS

In this Appendix, we provide further experimental details and results for the experiments in Sections 3 and 4.

Table 5: MLP architectures used in the experiment.

| Model | Taylor-Green vortex | Kovasznay flow | Acoustics | Euler gas | MaghetoHydroDynamics |
|---|---|---|---|---|---|
| **PINN** | | | | | |
| Layers and units | 2x64 | 4x64 | 4x64 | 8x512 | 8x256 |
| Activation | Tanh | Tanh | Tanh | Tanh | Tanh |
| **NCL** | | | | | |
| Layers and units | 2x64 | 4x64 | 4x64 | 8x512 | 8x256 |
| Activation | SoftMax | SoftMax | SoftMax | SoftMax | SoftMax |
| **CP-PINN modules** | | | | | |
| Layers and units | 2x64 | 4x64 | 4x64 | 8x256 | 8x256 |
| Activation | Tanh | Tanh | Tanh | Tanh | Tanh |
| **CP-NCL modules** | | | | | |
| Layers and units | 2x64 | 4x64 | 4x64 | 8x256 | 8x256 |
| Activation | SoftMax | SoftMax | SoftMax | SoftMax | SoftMax |

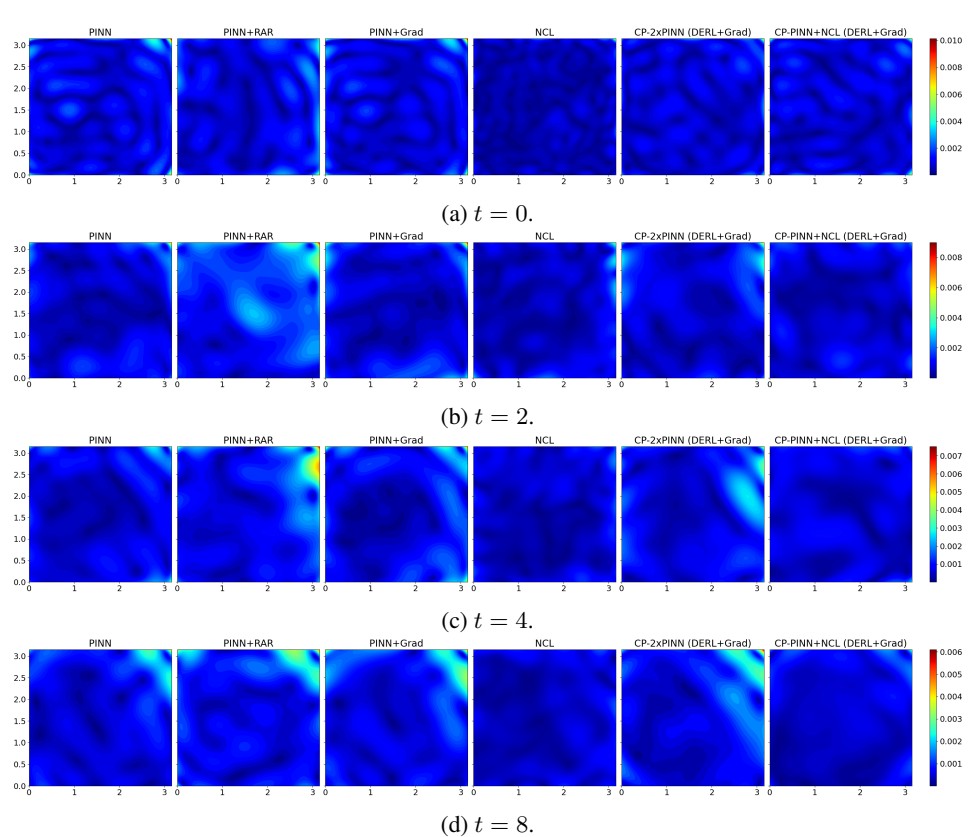

(a) $t = 0$.

(b) $t = 2$.

(c) $t = 4$.

(d) $t = 8$.

Figure 6: Taylor-Green vortex. Model pointwise error for the solution $(\boldsymbol{u}, p)$. Only the best alignment for each module choice is shown.

## D.1 TAYLOR-GREEN VORTEX

The analytical solution of the Taylor-Green vortex (Chorin, 1968) used in our experiment is given by

$$\boldsymbol{u}(x, y, t) = [\sin x \cos y, -\cos x \sin y]^\top e^{-2\nu t}, \quad p(x, y, t) = \frac{1}{4}(\cos(2x) + \cos(2y))e^{-4\nu t}. \quad (29)$$

Figure 6 shows the model errors at different times.

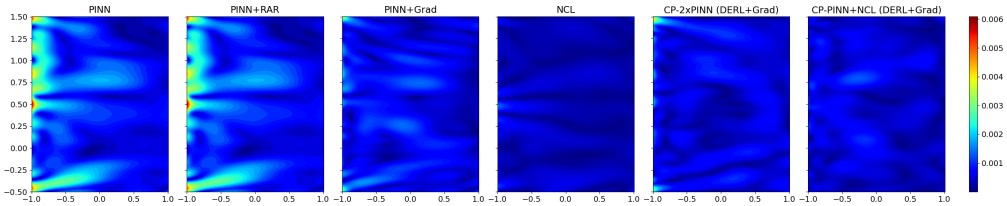

Figure 7: Kovasznay flow experiment. Prediction errors for PINN (and its variations), NCL, and CP models (one for each combination of modules). The lowest errors are in blue.

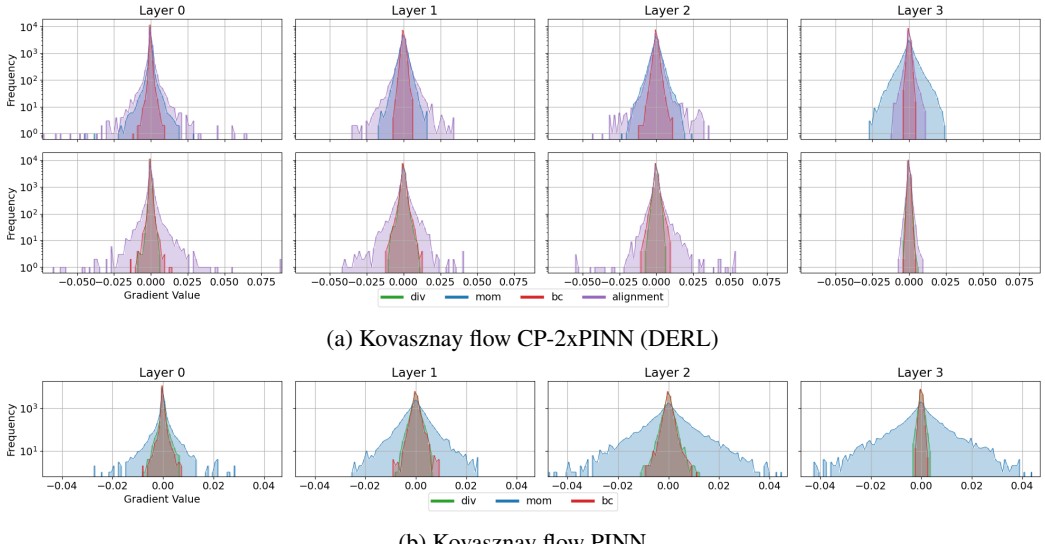

Figure 8: Gradient histograms. Each plot contains the histograms for the distribution of the gradients propagated at each layer of CP and PINN at the beginning of training, similarly to Wang et al. (2021).

## D.2  KOVASZNAY FLOW

The analytical solution to the Kovasznay flow (Drazin & Riley, 2009) is given by

$$\boldsymbol{u}(x,y) = \left[1 - e^{\lambda x}\cos(2\pi y), \frac{\lambda}{2\pi}e^{\lambda x}\sin(2\pi y)\right]^{\top}, \qquad p(x,y) = \frac{1}{2}\left(1 - e^{2\lambda x}\right), \qquad (30)$$

where $\lambda = \frac{1}{2\nu} - \sqrt{\frac{1}{4\nu^2} + 4\pi^2}$. In this case, the vorticity is given by $\omega = \frac{\lambda}{\nu}e^{\lambda x}\frac{\sin(2\pi y)}{2\pi}$. Figure 7 shows the errors of the employed models in the domain.

**Gradient Analysis.** Here, we provide a similar analysis to the one in Section 3.4 for the Kovasznay flow experiment, comprising 2 equations and 2 modules. Figures 8a and 8b show the distribution of the propagated gradients in a training step at the different layers of the MLP, respectively for the CP-2xPINN (each row is a module) and PINN models. The gradients for each loss term are represented with a different color. We clearly see the effect of learning the system in a compositional manner: the distribution histograms of the gradients of the loss terms are much more aligned than in the PINN model. This means that each loss can be learned more effectively without additional regularization (Wang et al., 2021).

## D.3  ACOUSTICS EQUATIONS

The initial condition of the Acoustics experiment in Section 3.3 is given by

$$p(0,x,y) = 1 + \cos\left(\frac{\pi}{0.2}(\sqrt{x^2 + y^2} - 0.5)\right), \qquad |\sqrt{x^2 + y^2} - 0.5| < 2, \qquad (31)$$

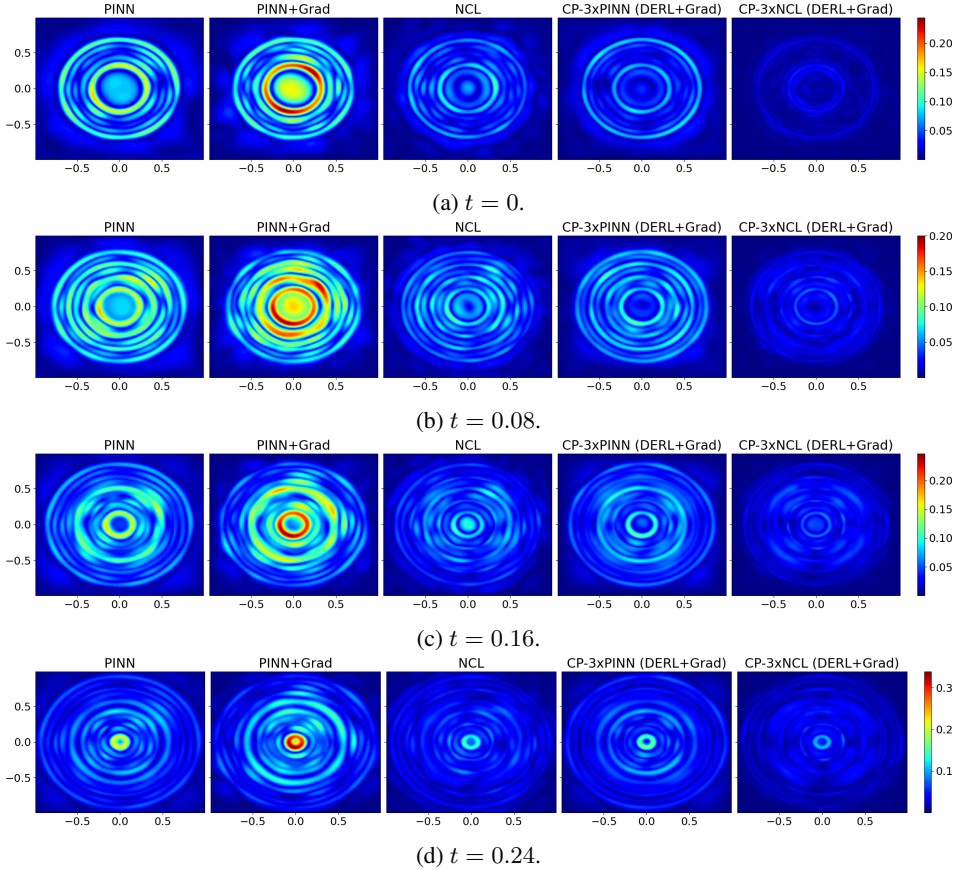

Figure 9: Acoustics equations. Model pointwise error for the solution $(p, u, f)$. Only the best alignment for each module choice is shown.

and null velocities in the whole square. Figure 9 shows the model errors at different times.

### D.4 NS-EULER GAS EQUATIONS

We provide further details on the setup of the experiment in Section 4.1. The initial conditions are given by

$$\rho_0(x, y) = (\sin(2\pi x) + \sin(2\pi y))^2 + 1, \qquad \boldsymbol{u}_0(x, y) = \left[e^{\sin(2\pi y)}, e^{\sin(2\pi x)}/2\right]^\top, \qquad (32)$$

To embed points from the 2D unit square $[0, 1]^2$ to the two-dimensional torus $\mathbb{T}$, the following function is used:

$$i : [0, 1]^2 \to \mathbb{T}^2, \qquad i(x, y) = (\cos(2\pi x), \sin(2\pi x), \cos(2\pi y), \sin(2\pi y)), \qquad (33)$$

which also automatically imposes periodic boundary conditions on the square.

Figure 10 shows the model errors at different times. We can see that the PINN and NCL models suffer in certain regions of the domain, while our models do not. In this case, Gradient-based reweighting worsens the performance of PINNs.

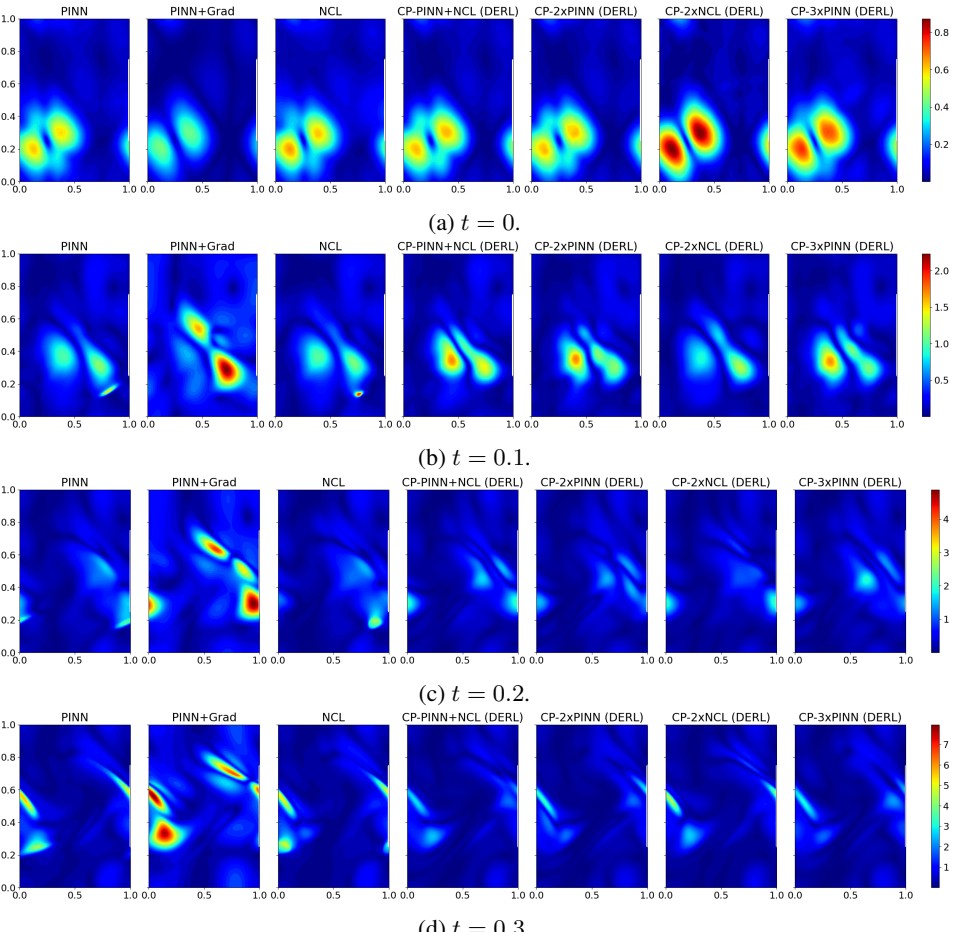

Figure 10: NS-Euler gas equations. Model pointwise error for the solution $(\rho, \boldsymbol{u}, p)$. Only the best alignment for each module choice is shown.

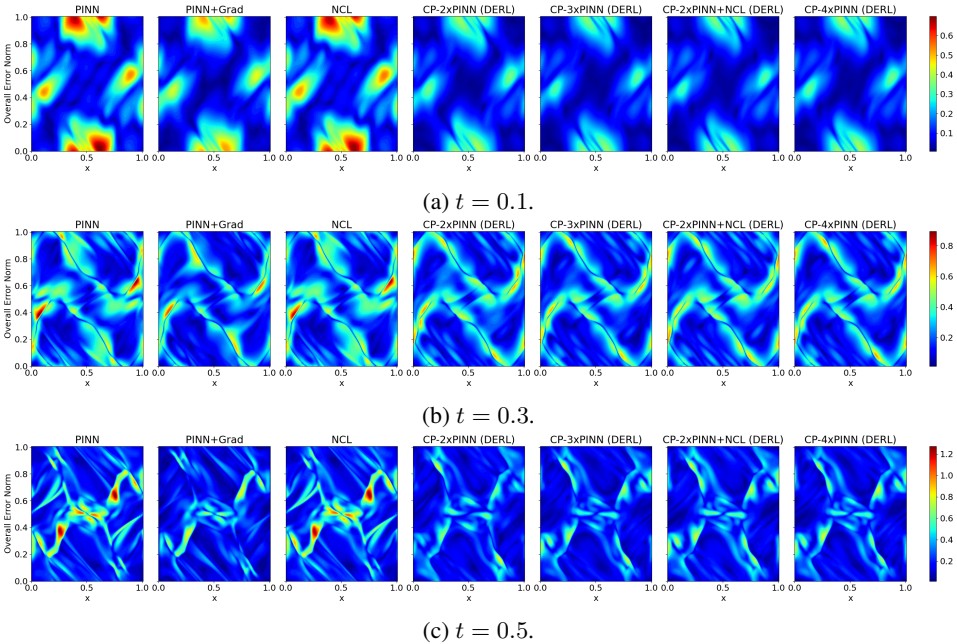

(a) $t = 0.1$.

(b) $t = 0.3$.

(c) $t = 0.5$.

Figure 11: MHD equations. Model pointwise error for the solution.

### D.5 MAGNETOHYDRODYNAMICS

Here, we provide additional details on the experiment in Section 4.2. The initial condition is given by

$$
\begin{aligned}
\rho(0, x, y) &= \frac{\gamma^2}{4\pi} \\
\boldsymbol{u}(0, x, y) &= [-\sin(2\pi y), \sin(2\pi x)] \\
P(0, x, y) &= \frac{\gamma}{4\pi} \\
A(0, x, y) &= \frac{\cos(4\pi x)}{4\pi\sqrt{4\pi}} + \frac{\cos(2\pi y)}{2\pi\sqrt{4\pi}} \\
\boldsymbol{B}(0, x, y) &= \nabla \times A
\end{aligned}
\tag{34}
$$

where $A(t, x, y)$ is the magnetic potential (Gruber & Rappaz, 1985). Similar to the NS-Euler experiment in Section 4.1, to satisfy the periodic boundary condition, we embed the square domain $[0, 1] \times [0, 1]$ in the two-dimensional torus $\mathbb{T}$ with equation 33 from Appendix D.4. The data generation is done as in Gopakumar et al. (2025).

Figure 11 shows the prediction errors of the models in the domain at different times in $[0, 0.5]$.

### E FURTHER ANALYSIS ON THE ALIGNMENT MECHANISM

**Loss curves.** We start by showing how the alignment loss, the prediction error, and the PDE residuals for transferred constraints are related to each other in practical examples. Figure 12 reports smoothed loss curves for the CP models on the Acoustic equation experiment of Section 3.3. These losses are calculated on the inference module, which learns equation (A.P) in table 1, but never learns equations (A.Vx) and (A.Vy) directly. These PDEs are learned by the other two modules. The inference module learns to satisfy these constraints only by aligning with the other two modules. Figure 13 shows smoothed loss curves for the CP models on the Kovasznay flow experiment in a similar manner.

We now investigate how well the modules are aligned during training by keeping track of their prediction errors for the common variables during the whole training process. This shows us how

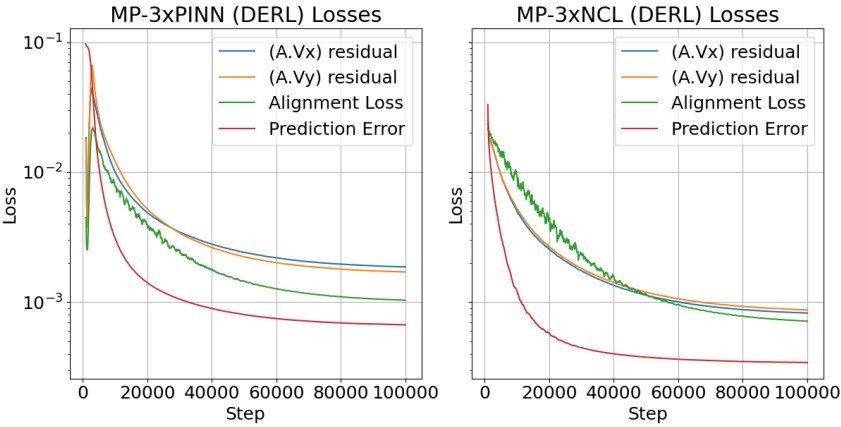

Figure 12: Smoothed loss curves of CP models on the Acoustic equation experiments. Losses are calculated on the inference module, which never learns equations (A.Vx) and (A.Vy) directly.

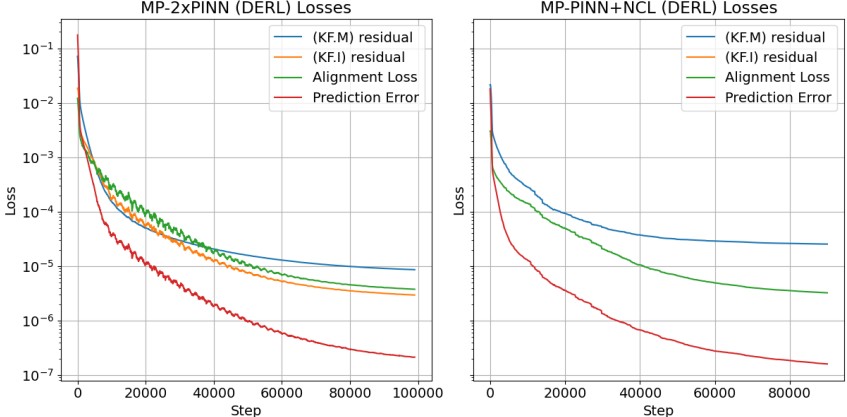

Figure 13: Smoothed loss curves of CP models on the Kovasznay flow experiment. Losses are calculated on the inference module, which never learns equation (KF.I) directly.

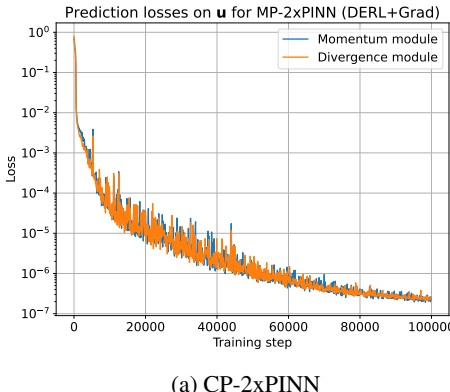
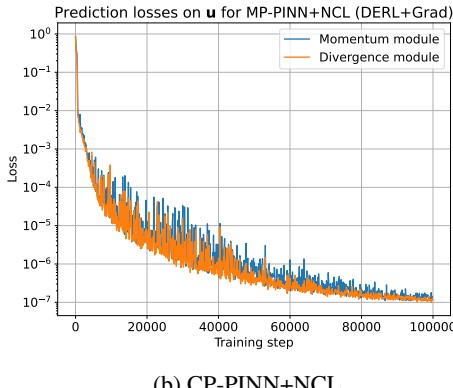

(a) CP-2xPINN                    (b) CP-PINN+NCL

Figure 14: Prediction errors on the shared variables for the CP modules on the Kovasznay flow experiment

well their predictions are aligned during the whole process. If the alignment mechanism is working as desired, the loss curves should be very similar, meaning that the two modules are working as one and learning together. For the Kovasznay flow experiment, we report the errors for $u$ for the Momentum and divergence modules in Figure 14, for both the CP-2xPINN and CP-PINN+NCL models with DERL alignment and gradient-based reweighting. For the Acoustic equations experiment, we have that the pressure and velocity x modules share the variables $(p, u)$, while the pressure and velocity y modules share $(p, v)$. Hence, we have three plots: one shows the prediction errors for all three modules on $p$, one shows the prediction errors for the first and second module on $(p, u)$, and the third shows the precision errors for the first and third module on $(p, v)$. These results are available in Figure 15 for both the CP-3xPINN and CP-3xNCL models with DERL alignment and gradient-based reweighting.

These plots clearly indicate that the losses are very similar throughout the whole training process, indicating that the alignment mechanism is making the modules collaborate closely to obtain the final solution.

**Measuring discrepancies.**   We now show from a numerical point of view how close the functions predicted by different modules are in practical terms. We perform an additional test on the Kovasznay Flow and Taylor-Green experiments from Section 3. At the end of training, we predict the solution for the variable $u$, which is a vector containing the x and y components of the fluid velocity, shared among the two modules of the CP models. We measure the following quantities:

- The $L^2$ distance between the two modules, normalized with respect to the true solution norm $\frac{\|u_{\text{module 1}} - u_{\text{module 2}}\|_2}{\|u_{\text{true}}\|_2}$: this indicates how different are the two predicted solutions, scaled to the magnitude of the true function.

- For a point-wise comparison, we consider the symmetrized Percentage Error: for each point in the domain, we calculate $\frac{2\|u_{\text{module 1}}(t,x) - u_{\text{module 2}}(t,x)\|}{\|u_{\text{module 1}}(t,x)\| + \|u_{\text{module 2}}(t,x)\|}$ which measures how far are the two predictions compared to each other. We then calculate the mean and maximum value.

We perform this test for both the CP-2xPINN and CP-PINN+NCL models. Results are available in Table 6. From the scale of the metrics, we conclude that the difference between the two modules is negligible compared to the scale of the solution or with respect to each other. We have not observed any cases where the modules disagree with each other, as they are practically the same function when it comes to predicting the solution.

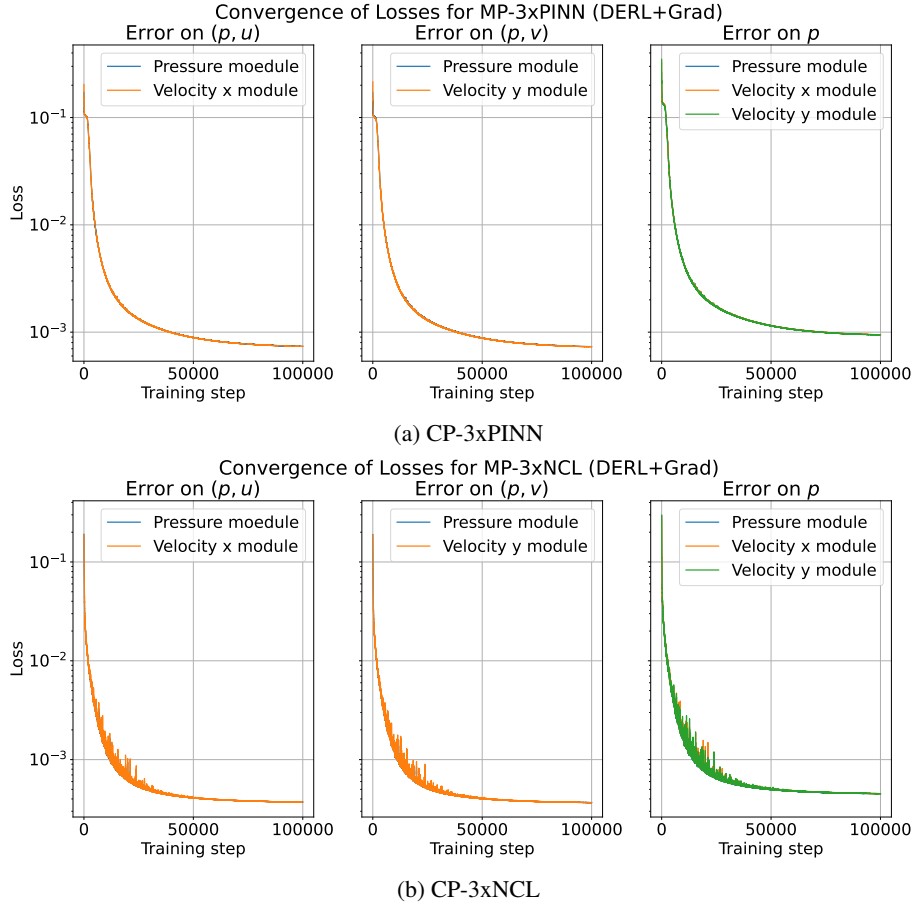

(a) CP-3xPINN

(b) CP-3xNCL

Figure 15: Prediction errors on the shared variables for the CP modules on the Acoustic equation experiment

Table 6: Normalized $L^2$ distance and symmetrized Percentage Error metrics for the modules on predicting $u$ in the Kovasznay flow and Taylor Green vortex experiments.

| Kovasznay Flow Experiment | | | |
|---|---|---|---|
| **Model** | **Normalized $L^2$ distance** | **Mean sPE** | **Max sPE** |
| | $\times 10^{-4}$ | $\times 10^{-7}$ | $\times 10^{-6}$ |
| **CP-2xPINN (DERL+Grad)** | 2.758 | 6.095 | 2.954 |
| **CP-PINN+NCL (DERL+Grad)** | 2.210 | 4.874 | 1.729 |
| **Taylor-Green vortex Experiment** | | | |
| **Model** | **Normalized $L^2$ distance** | **Mean sPE** | **Max sPE** |
| | $\times 10^{-4}$ | $\times 10^{-7}$ | $\times 10^{-6}$ |
| **CP-2xPINN (DERL+Grad)** | 8.910 | 2.762 | 2.102 |
| **CP-PINN+NCL(DERL+Grad)** | 10.47 | 3.187 | 4.693 |

Table 7: Results for different values of $\lambda_{\text{align}}$ on two CP models.

| Acoustic equations | | |
|---|---|---|
| **CP-3xNCL (DERL)** | $L^2$ **error** $\times 10^{-5}$ | **max error** $\times 10^{-1}$ |
| $\lambda_{\text{align}} = 1$ | 5.165 | 1.547 |
| $\lambda_{\text{align}} = 0.1$ | 3.164 | 1.257 |
| $\lambda_{\text{align}} = 0.01$ | 2.460 | 1.089 |
| $\lambda_{\text{align}} = 10.$ | 12.46 | 4.388 |

Table 8: Results for the Acoustic equation experiment where NCL and PINN have the same parameter budget of CP during training. In these cases, PINN and NCL have 3 times the number of parameters of CP during inference.

| Acoustic equations | | |
|---|---|---|
| **Model** | $L^2$ **error** | **max_err** |
| | $\times 10^{-5}$ | $\times 10^{-1}$ |
| **PINN** | 5.671 | 1.462 |
| **NCL** | 2.940 | 1.169 |
| **CP-3xNCL (DERL+Grad)** | **2.718** | **1.121** |

# F ABLATIONS

**Alignment coefficient $\lambda_{\text{align}}$.** We experimented with different values of $\lambda_{\text{align}}$ to analyze the sensitivity of CP with respect to this hyperparameter. We test values of $\lambda_{\text{align}}$ between $0.01$ and $10$. Results for the Acoustics experiment (CP-3xNCL model with DERL alignment) are available in table 7

As we can see, $\lambda_{\text{align}}$ regulates the importance of the alignment between models. While its influence depends on the specific task and modules, the $\lambda_{\text{align}} = 1$ works well on average in all benchmarks. Highly specialised tuning of the hyperparameter can lead to modest improvement, as in the acoustic equation case above.

**Number of parameters in the baselines.** Since the CP model employs more than one MLP during training, the total number of parameters during this phase is increased. At inference, where the models can be used indefinitely, CP uses a single module, and hence will have the same number of parameters and expressiveness as an NCL or PINN module. In this ablation, we increase the number of units in the NCL and PINN layers to match the parameter budget of the CP model during training. We experimented with the Acoustics equations and the CP model with 3 NCL modules. We remark that, during inference, this means that PINN and NCL will use 3 times the parameters of CP. Results are available in Table 8. As we can see, our models perform better even though they use fewer parameters during inference, further showing the importance of modularization in these applications.

**Statistical Significance.** To show robustness with respect to different initialization and stochasticity in optimization, we perform additional runs with different seeds on the Acoustic equations experiment. We experiment with the PINN and NCL baselines as well as the CP-3xNCL model, DERL alignment, and gradient-based reweighting. Results averaged over three seeds are available in Table 9.

As we can see, the results clearly indicate that our CP model with 3 NCL modules is still the best model by far, with lower $L^2$ and maximum errors in the domain. Furthermore, the standard deviations are smaller, indicating more robustness to different initializations.

# G FURTHER THEORETICAL DISCUSSION

In this Section, we provide additional theoretical understandings and intuitive insights on why our modular approach is easier to optimize compared to a PINN model.

Table 9: Results for the Acoustic equation experiments with different seeds. Results are averaged over 3 runs. We report mean and standard deviations.

| Acoustic equations | | |
|---|---|---|
| **Model** | $L^2$ **error** | **max_err** |
| | $\times 10^{-5}$ | $\times 10^{-1}$ |
| **PINN** | $5.090_{\pm 0.859}$ | $2.247_{\pm 0.413}$ |
| **NCL** | $3.540_{\pm 0.431}$ | $1.610_{\pm 0.082}$ |
| **CP-3xNCL (DERL)** | $1.723_{\pm 0.173}$ | $1.091_{\pm 0.088}$ |

**On the difference between PDE residual terms and alignment losses**  A PDE residual term imposes constraints that link together the partial derivatives of an MLP. For example, a PDE such as $\frac{\partial u}{\partial t} - \lambda \cdot \nabla u = 0$ imposes that the time derivative of the network must be equal to a linear combination of the spatial ones. As observed in Krishnapriyan et al. (2021), this type of constraint is often ill-conditioned and can lead to numerical instabilities. The same work also shows that these PDE terms can produce a very complex and non-smooth loss landscape. Hence, our idea is to avoid combining multiple PDE losses in a single MLP. ComPhy tackles subproblems (simpler than the whole system) in each model, and then transfers information across models through alignment.

The alignment loss, on the other hand, does not link together the partial derivatives or the outputs of an MLP. Instead, it uses the derivatives or outputs of one module as a target for those of another module in a supervised manner (to be precise, this is similar to a distillation between the two models). Hence, we expect this type of loss to reduce the complexity of the problem and to simplify the optimization process.

**Physical constraints and properly aligned modules.**  When two models are properly aligned, that is, when the SOB or DERL alignment loss is close to zero, the functions predicted by the modules are very close in the Sobolev $W^{1,2}$ space (Trenta et al., 2026). As an example, we use the one in Section 2.3 on the Navier-Stokes equations. After training, module 1 has learned the momentum equation, while module 2 has learned the divergence equation. Since $\boldsymbol{u}_{\text{module 1}} \approx \boldsymbol{u}_{\text{module 2}}$ and $D\boldsymbol{u}_{\text{module 1}} \approx D\boldsymbol{u}_{\text{module 2}}$, if we calculate the divergence of module 1, we can say that $\nabla \cdot \boldsymbol{u}_{\text{module 1}} \approx \nabla \cdot \boldsymbol{u}_{\text{module 2}} \approx 0$. This means that a correct alignment allows module 1 to satisfy the divergence equation because module 2 does so. This intuitively provides a reason why module 1 is still capable of learning the system while seeing only one equation. We experimentally validated this claim in Appendix E.

