# OpenReview forum: "ComPhy: Composing Physical Models with end-to-end Alignment"
_ICLR.cc/2026/Conference — ICLR 2026 Poster_

### Official Review · Reviewer_M37D · 2025-10-26

**Soundness:** 1
**Presentation:** 1
**Contribution:** 2
**Rating:** 2
**Confidence:** 4

**Summary:**

This paper presents a modular framework to tackle the optimization problem of PINNs caused by multiple loss constraints, which is named ComPhy. Specifically, ComPhy proposes to split multiple loss functions into several subsets with shared physical quantities. Then, several modules are configured as PINN or NCL models, and each module will be optimized based on one subset of loss functions. Besides, an alignment loss is newly proposed to align the shared physical quantities in different modules. As for inference, ComPhy only needs to infer the module with complete quantities once. Such a modular framework is expected to ease the difficulty in joint optimization of multiple losses and ensure the physical alignment in the final results. The authors provide sufficient experiments to verify the effectiveness of the proposed ComPhy.

**Strengths:**

(1)	I think the proposed idea is interesting and novel, especially in assigning different loss subsets to different modules.

(2)	The experiments are sufficient to deliver a comprehensive evaluation of the proposed method.

(3)	Rich implementation details are included.

**Weaknesses:**

### (1) The motivation of ComPhy. Why does this design work?

I think the authors fail to elaborate on why ComPhy performs well. The only statement is in Lines 99-101, that is, “State-of-the-art models like PINNs may suffer from optimization problems when multiple PDEs are involved. ComPhy avoids this issue by using different modules to optimize the different PDEs of the system separately.” However, suppose one of the modules in ComPhy contains complete physics quantities. In that case, it will also be optimized by the newly added alignment losses, which is still a multiple-loss optimization problem. Why is the module optimized in this way better than the model optimized from multiple PDE losses? I think an intuitive understanding is required.

Besides, let us consider the example in Section 2.3. If ComPhy employs two PINN models as modules, at the beginning of training, it is really hard for the second module optimized with Eq.~(8) to generate a reliable solution. Why can the alignment loss help the first module be optimized better?

Therefore, I cannot understand why ComPhy can help with the training. **Maybe the visualization of training curves (training, alignment and test losses) can be a good choice for elaboration. In addition, some theoretical analyses or thought experiments are expected.**

### (2) Too many unjustified or vague claims.

I think this paper contains many unsupported claims or statements, which seriously damage the scientific rigor. Here are some examples:

-	Abstract: “CP is the first approach specifically designed to tackle systems of PDEs”. Suppose the authors refer to “systems of PDEs” as the combination of multiple PDE equations. In that case, I think there are many related works that tackle the optimization problem of balancing multiple PINN losses, such as [1].

-	Line 99: “State-of-the-art models like PINNs may suffer from optimization problems when multiple PDEs are involved”. What kind of “optimization problems” do you mean? I think if the authors cannot detail this statement, the motivation of this paper is unclear.

-	Line 111: “The solution to a PDE system is unique only if all the PDEs are satisfied at once (Evans, 2022).” Although this is not a core statement, it should be noted that many PDEs contain multiple solutions.

-	Eq. (3): The authors do not provide a clear definition for the L_{align}, since in Eq. (3), each row defines one type of L_{align}. I do not know what the final version used in Eq. (4) is.

-	Table 1: It is really hard to understand the last column. After a long time thinking, I understand that each row of the last column represents one configuration in ComPhy. I think a more direct description is required.

-	All the numbers, like 100.000 or 600.000, should be 100,000 or 600,000.

-	Line 356: “The authors show that when PINNs are optimized correctly, the gradients tend to be evenly distributed across all layers.” This claim is not correct, since in Wang et al. (2021), the main focus is on the imbalance among different losses. Thus, this statement should be “the gradients of multiple PDEs”.

[1] Wang et al. When and why pinns fail to train: A neural tangent kernel perspective. Journal of Computational Physics, 2022.

### (3) How to decide the configuration of ComPhy, such as pure PINNs, pure NCLs, or a combination of PINN and NCL?

As presented in Table 2, different configurations lead to quite different results. When using ComPhy, it may take a long time to tune the concrete configuration of the modular framework.

###  (4) About related work.

I think this paper is not related to the neural operator, which is purely data-driven. The authors should spend more time reviewing papers about PINN optimization, such as [1,2,3].

[1] Wang et al. When and why pinns fail to train: A neural tangent kernel perspective. Journal of Computational Physics, 2022.

[2] Daw et al. Mitigating propagation failures in physics-informed neural networks using retain-resample-release (r3) sampling, ICML 2023.

[3] Wu et al. RoPINN: Region Optimized Physics-Informed Neural Networks, NeurIPS 2024.

**Questions:**

Please see Weaknesses. To highlight, I think the authors should answer the following questions carefully:

-	Why does ComPhy work well?

-	How to decide the configuration of ComPhy?

---

> ### Author Response · Authors · 2025-11-19
> **Rebuttal (part 1/3)**
>
> We thank the reviewer for their comments and valuable feedback, which is important to improve the quality of our work. We also thank the reviewer for highlighting the novelty of our idea and the quality of our experiments and implementation details.
>
> We now address the specific comments raised by the reviewer:
> ## W1: Motivation and why ComPhy works.
> First, we remark on the differences between a PDE residual term and the alignment loss proposed.
> - In a PDE residual term, the partial derivatives of the MLP are entangled together in a single loss term. For example, a PDE such as $\partial u/\partial t - \lambda\cdot \nabla u$ imposes that the time derivative of the network must be equal to a linear combination of the spatial ones. As observed in [1], this type of constraint is often ill-conditioned and can lead to numerical instabilities. The same paper also shows that these PDE terms can produce a very complex and non-smooth loss landscape.
> - On the other hand, the alignment loss uses the derivatives of module 2 as a target to those of module 1 in a supervised manner (to be precise, this is similar to a distillation between the two models). We expect this kind of loss to produce an easier optimization problem for the MLP, which has one less PDE residual term to optimize.
>
> We also recall the experimental validation in Section 3.4, which was specifically performed to analyze the benefit of our modular approaches and the alignment mechanism. The results are clear: the gradients of the different loss terms are more evenly distributed. This, the different nature of the two losses, and the smaller number of losses in a single module, all allow the ComPhy to learn the constraints faster and more stably.
>
> About the example in Section 2.3, the performance at the beginning of training is not relevant, as the two modules are not aligned, nor have they learned to satisfy each PDE constraint. We remark that the alignment mechanism, one of our main contributions, is designed specifically for avoiding conflicts and for allowing ComPhy and its modules to provide reliable solutions. As we discussed in Section 2.4, we are interested in when the modules are properly aligned. In this case, if the alignment loss (either SOB or DERL) is close to zero, modules 1 and 2 predict functions that are very close to each other in the Sobolev space $W^{1,2}$. At the end of training, module 1 has learned the momentum equation, while module 2 has learned the divergence equation. Since $u_{\text{module 1}} \approx u_{\text{module 2}}$ and $Du_{\text{module 1}} \approx Du_{\text{module 2}}$. If we calculate the divergence of module 1, we can say that $\nabla\cdot u_{\text{module 1}} \approx \nabla \cdot u_{\text{module 2}}\approx 0$. The alignment allowed for module 1 to satisfy the divergence equation even though it was not a constraint directly imposed on it, but because module 2 does so.
> We added these discussions to the main text and appendix of our revised paper to improve clarity.
>
> For further visualization of the effect of the alignment on the prediction error, in the revised version of the paper (Appendix E), we provide additional loss curves (smoothed for clarity) for the acoustic equation and the Kovasznay flow experiments for different ComPhy models. In each case, we calculate the following losses on the inference module:
> - the alignment loss
> - the prediction loss,
> - The PDE residuals of the PDEs that are NOT learned by the inference module (but which should be learned through the other ones)
>
> The last one indicates how well the inference module satisfies the PDE residuals that it did not learn directly. These PDE constraints are satisfied through the alignment with the other modules, which learned them. As we can see, the losses are nicely correlated, which indicates that the better the alignment, the better the PDE constraints are transferred and the better the performance of the module.
>
> [1] Krishnapriyan et al. Characterizing possible failure modes in physics-informed neural networks. NeurIPS 2021

---

> > ### Author Response · Authors · 2025-11-19
> > **Rebuttal (part 2/3)**
> >
> > ## W2: Claims.
> > We respond point by point:
> > - We acknowledge the fact that prior works tackled systems of PDEs by balancing loss terms in a single PINN. However, we are the first to exploit the inductive bias given by a system of PDEs, where multiple equation constraints have to be satisfied at once. Existing methodologies treat the loss terms as one monolithic block, for example, by reweighting the terms based on their gradient norms. Instead, we adopt an elegant and modular approach designed to combine the interwoven dynamics of multiple PDEs. We will revise this sentence in the paper to better specify this difference.
> > - We included references to the optimization problems we were referring to both in Section 2.1 and in Section 5. In particular, [1] shows that the PDE residual terms are often ill-conditioned, leading to numerical instabilities, while [2] shows that PDE loss terms can produce imbalanced gradients in the layers of the MLPs. These are only two examples, and our related works section reports more relevant works on known issues in PINN training, along with their characterization. We will expand our methodology and related works sections with a more in-depth discussion of these topics for better clarity.
> > - We understand the concern of the reviewer about our statement on the uniqueness of solutions of PDEs, and we will correct the wording in a revised version of the paper. We know that a PDE can have multiple solutions. What we cared to express with that sentence is that in a system of multiple PDEs, one equation is not enough to produce the solution (in fact, with just one PDE, multiple solutions are possible), but all three are necessary to obtain the true (and possibly unique) one. Hence, the requirement for an alignment between the modules is justified.
> > - In equation 3, we indicate three different possibilities for the alignment losses: each of them is a valid option to be used in equation 4. In Section 2.4, we also provide some claims on which of them we expect to work better. In our experimental results (Tables 2 and 3), the reviewer can see that we include a specific column for the alignment used in that CP model, and we try all of them in our case studies, to search for the optimal one. Our results show that the OUTL alignment is the worst, as expected, while the other two perform similarly, with a slight preference for DERL. Hence, we concluded that DERL was the best choice in general. The influence of the particular alignment loss is discussed throughout both Sections 3 and 4 in our experimental results.
> > - We will improve the clarity of the last column in the revised version of the paper, specifying that the last column indicates the combinations of modules we tested in ComPhy.
> > - We revised the notations for numbers over one thousand in the updated version of the paper.
> > - We thank the reviewer for the suggestion to clarify the claim in line 356. The meaning is the same, but we implicitly assumed we were discussing the gradients of multiple PDEs. Our analysis remains correct and consistent, as the Figures clearly indicate that the gradients of the different loss terms are evenly distributed within each layer.
> >
> > [1] Krishnapriyan et al. Characterizing possible failure modes in physics-informed neural networks. NeurIPS 2021
> > [2] Wang et al. When and why pinns fail to train: A neural tangent kernel perspective. Journal of Computational Physics, 2022.
> >
> > ## W3: How to decide the configuration of ComPhy.
> >
> > We agree with the reviewer that the choice of the modules is relevant for the best performance of the overall model. In our experiments, apart from the MHD one, where we tested the scaling capabilities of ComPhy, we explored various combinations to demonstrate the flexibility and robustness of ComPhy to different choices of learning modules. Depending on the system of PDEs, we tried every possible combination of modules, given that the NCL module requires a divergence-free equation. Our results support the fact that, when possible, a combination of modules that includes NCL is always beneficial. From a practical point of view, we suggest using NCL modules when there is one or more divergence-free equations. Similarly, since most physical systems present 2 or 3 equations, the most common and effective choice is to use 2 or 3 modules, respectively. We included this discussion at the end of Section 4 in our paper.

---

> > > ### Author Response · Authors · 2025-11-19
> > > **Rebuttal (part 3/3)**
> > >
> > > ## W4: On the related works.
> > > We thank the reviewer for pointing out additional interesting works in this field of research, which we discuss in the revised manuscript. While [1] has already been discussed, we make additional comments on [2] and [3].
> > >
> > > [2] proposes a well-thought-out resampling strategy to adaptively choose points where PINNs are trained. Regions with higher residuals are more likely to be chosen, while regions with lower residuals can be limited, but it is important to have representatives from each of them.
> > >
> > > [3] modifies the PDE residual terms by using a variational formulation in small regions around collocation points to reduce the generalization error
> > >
> > > Finally, we included the neural operator models in our related works due to their recent popularity. For a broader audience, we already discussed the main differences between our setting and theirs.
> > >
> > > References:
> > > [1] Wang et al. When and why pinns fail to train: A neural tangent kernel perspective. Journal of Computational Physics, 2022.
> > > [2] Daw et al. Mitigating propagation failures in physics-informed neural networks using retain-resample-release (r3) sampling, ICML 2023.
> > > [3] Wu et al. RoPINN: Region Optimized Physics-Informed Neural Networks, NeurIPS 2024.
> > >
> > >
> > > ## Questions:
> > > We answer question 1 in W1 and question 2 in W3.

---

> > > > ### Comment · Reviewer_M37D · 2025-11-27
> > > >
> > > > I would like to thank the authors for the detailed rebuttal. Most of my concerns have been resolved. Thus, I decided to raise my score to 6.
> > > >
> > > > In general, I like the idea of ComPhy. The explanation about model distillation sounds reasonable, and the training curve is also included. I think the authors should conduct more investigations into the collaboration and convergence among different sub-models. That will be helpful to clarify how ComPhy works during training. For example, will different models converge at the same time, or which one will converge first?

---

> > > > > ### Author Response · Authors · 2025-11-28
> > > > >
> > > > > We thank the reviewer for further engaging in the discussion, for the rise in the score, and their appreciation of our work. We are delighted to see that our effort to provide clarifications and further insights on our model has been appreciated by the reviewer. We also think that the concerns they raised helped us improve our work.
> > > > >
> > > > > As suggested, we further investigated the convergence of the different modules during training. To do so, we kept track of their prediction loss with respect to the ground truth. Since different modules may not predict the entire set of variables, we considered only the common ones. In particular:
> > > > > - For the Kovasznay flow experiment, both modules predict $\mathbf{u}$, the velocity field. Thus, we keep track of the prediction error for this variable during training for both modules.
> > > > > - For the Acoustics equations experiment, one module predicts $(p,u,v)$, while the others predict respectively $(p,u)$ and $(p,v)$. For a complete comparison, we compare the loss curves of $p$ for all three modules, the loss curves of the first and second on $(p,u)$ (their common variables), and on $(p,v)$ for the first and third.
> > > > >
> > > > > We report these loss curves in Appendix E (Figures 14 and 15, and the second paragraph in Appendix E). As we can see, the modules have very similar losses during the whole training process. Thus, the alignment mechanism is working as desired, making the different modules behave very similarly to each other. We think this result further strengthens our claims on the alignment.
> > > > >
> > > > > We kindly ask the reviewer if they are willing to raise the score once more, if no other outstanding concerns remain.

---

### Official Review · Reviewer_Jj5v · 2025-10-26

**Soundness:** 3
**Presentation:** 2
**Contribution:** 3
**Rating:** 6
**Confidence:** 3

**Summary:**

The paper introduces ComPhy, a framework for solving systems of PDEs. Instead of training a single monolithic model on all equations, ComPhy assigns each PDE to a dedicated learning module (e.g., a PINN or NCL) and enforces alignment losses between modules that share physical variables. These alignment losses encourage consistency across modules and improve convergence.

**Strengths:**

1. The paper introduces a novel modular design. The decomposition of multi-PDE systems into specialized modules is elegant and well-motivated both computationally and physically.

2. Introducing Sobolev-inspired alignment effectively transfers physical information between modules and leads to empirical gains.

3. Gradient distribution studies convincingly explain why ComPhy’s modular approach stabilizes training compared to conventional PINNs.

**Weaknesses:**

See questions below.

**Questions:**

1. Managing multiple interacting modules may increase computational and memory overhead, particularly for systems with many PDEs. It would be helpful if the authors could clarify how they address this issue.

2. Since the overall objective combines both alignment losses and module-specific PDE/BC/IC losses, it would be useful to report how sensitive the method is to the relative weighting of these terms. Does performance degrade significantly if the alignment coefficient is varied?

3. Can the modular design generalize to systems where PDEs share only partial or implicit variables?

4. The figures, particularly Figure 1, could be made more intuitive and easier to interpret.

---

> ### Author Response · Authors · 2025-11-19
> **Rebuttal**
>
> We thank the reviewer for the comments on the positive feedback on the modular design and empirical analysis, as well as the importance of a correct information transfer.
>
> ## Q1: Computational Efficiency.
> We understand the concern about the increased computational and memory overhead. In Appendix C.2, we provide the computational time of the different methods, showing that the ComPhy models are still very efficient even with multiple modules. In particular, depending on the task, the NCL model alone often scales worse than an NCL module in our ComPhy, as we do not need to propagate gradients for second-order derivatives through its particular architecture. Our model simplifies this, as an NCL module in ComPhy does not need to optimize a PINN loss.
>
> We now address the question of possible differences in performance with the same parameter budget during training. During inference, ComPhy uses only one module to predict the solution, and, in practical applications, we would expect the training phase to be performed once, while inference is performed multiple times. Given the fact that these models are not prohibitively large in the number of parameters (all models have between 20k and 500k), doubling the size of the model does not make a huge difference during training, while the cost during inference could accumulate.
>
> To further investigate this topic, we trained PINN and NCL models with additional units to match the same parameter budget during training for the Acoustic equation experiment. We remark that, in this case, ComPhy models will use fewer parameters during inference (one third, to be precise).
>
> | **Acoustic equations** |  |  |
> |:---:|:---:|:---:|
> | **Model** | **$L^2$ error** | **max_err** |
> |  | $\times 10^{-5}$ | $\times 10^{-1}$ |
> | **PINN** | 5.671 | 1.462 |
> | **NCL** | 2.940 | 1.169 |
> | **CP-3xNCL (DERL+Grad)** | **2.718** | **1.121** |
>
> In this case, even with more parameters at inference, the performance improvement of PINN and NCL is limited, with ComPhy performing better with fewer parameters.
> We added these results to our Ablations in Appendix F.
>
> As a final remark, since most physical systems involve 2 or 3 equations in the system of PDEs, and given the dimensions of our models, increasing the number of parameters by 2 or 3 times is totally reasonable, especially given the performance improvements.
>
> ## Q2: Hyperpatamers
> We understand the concern about the choice of the loss weights. This choice of the hyperparameters is task-specific, as for all the hyperparameters in PINN training, but (1) when used, gradient-based reweighting leads to the best performance and does not require choosing specific hyperparameters, alleviating the burden of model selection, and (2) a naive choice, such as $\lambda_{\text{align}}=1$, which we used in our experiments, works very well in all experiments, with possible improvements with particular choices this weight.
>
> We performed additional ablations on the value of $\lambda_{\text{align}}$ on the Acoustic equation experiments, which we report here:
> | **Acoustic equations** |  |  |
> |:---:|:---:|:---:|
> | **Model** | **$L^2$ error** | **max_err** |
> | CP-3xNCL (DERL) | $\times 10^{-5}$ | $\times 10^{-1}$ |
> | $\lambda_{\text{align}}=1$ | 5.165 | 1.547 |
> | $\lambda_{\text{align}}=0.1$ | 3.164 | 1.257 |
> | $\lambda_{\text{align}}=0.01$ | 2.460 | 1.089 |
> | $\lambda_{\text{align}}=10.$ | 12.46 | 4.388 |
>
> As we can see, $\lambda_{\text{align}}$ regulates the importance of the alignment between models. The $\lambda_{\text{align}}=1$ works well in our experiments, while highly specialised tuning of the hyperparameter can lead to modest improvement, as in this.
> We included this ablation in Appendix F of the revised manuscript.
>
> ## Q3: Shared variables.
> ComPhy can be applied to any system of PDEs and any set of variables with very minimal restrictions. We already apply ComPhy to systems of PDEs where some variables do not appear in each equation, and the modules align only the common variables with each other (Navier-Stokes, Acoustics, and MHD equations are some examples). In general, the minimal requirement to apply ComPhy is that each equation should share a variable with at least another equation in the system. If $v$ (explicit or implicit) is such a variable, then it is sufficient to add a corresponding term in the alignment loss.
> We are also happy to discuss any particular example of a physical system that the reviewer wants to further investigate to test ComPhy's applicability.
>
> ## Q4: Figures.
> We will improve the clarity and intuitiveness of Figure 1 in a revised version of the paper. We are also happy to include any particular suggestions by the reviewer to improve our figures.

---

### Official Review · Reviewer_AyTi · 2025-10-28

**Soundness:** 2
**Presentation:** 2
**Contribution:** 2
**Rating:** 4
**Confidence:** 3

**Summary:**

This paper presents ComPhy, a novel modular framework designed to leverage the inherent physical structure of the problem to solve systems of PDEs. ComPhy assigns each equation a dedicated learning module, like PINN or NCL, and introduces an end-to-end alignment mechanism to enable knowledge transfer between different modules. The results show that it outperforms state-of-the-art approaches where a single model is trained on all PDEs at once.

**Strengths:**

- From my perspective, this is a novel method designed for solving systems of PDEs. It is an interesting research field which have not been explored.
- The proposed method is novel and seems elegant for solving systems of PDEs. The experiment results also demonstrate that it outperforms plain PINNs.
- The paper is well written and easy to follow.

**Weaknesses:**

- It may be hard, but some theoretical understandings, even intuitive ones, could make the method more convincing.
- After training, we can use a subset of the trained networks to predict all physical variables. Can some networks sharing the same variables have conflicts?
- The paper lacks analysis on the efficiency of the model. For a system of N PDEs, each network requires (N+2) loss terms, and the whole system requires N*(N+2) loss terms; how does it influence the training time compared to plain PINNs.
- Can we replace the current PINNs and NCLs with neural operators? The current framework does not seem to support such modules.

**Questions:**

See weaknesses.

---

> ### Author Response · Authors · 2025-11-19
> **Rebuttal (part 1/2)**
>
> We thank the reviewer for their feedback and suggestions and for highlighting the novelty and elegance of our approach, as well as the clarity of the paper.
>
> We now address the specific weaknesses and questions raised by the reviewer.
>
> ## W1: Theoretical Understanding.
> We remark that proper theoretical statements and theorems in the PINN literature are hard to find (see, notably [2,3]), especially for systems of PDEs, for which we do not know of. Here, we try to provide an additional intuitive understanding of why ComPhy performs better than PINNs and the other baselines. First, we remark on the differences between a PDE residual term and the alignment loss proposed.
> - A PDE residual term imposes constraints that link together the partial derivatives of the MLP. For example, a PDE such as $\partial u/\partial t - \lambda\cdot \nabla u$ imposes that the time derivative of the network must be equal to a linear combination of the spatial ones. As observed in [1], this type of constraint is often ill-conditioned and can lead to numerical instabilities. The same paper also shows that these PDE terms can produce a very complex and non-smooth loss landscape. Hence, our idea is to avoid as much as possible to combine multiple PDE losses in a single MLP. ComPhy tackles subproblems (simpler than the whole system) in each model, and then transfers information across models through alignment.
> - On the other hand, the alignment loss does not link together the partial derivatives of the network. Instead, it uses the derivatives of one module as a target for those of another module in a supervised manner (to be precise, this is similar to a distillation between the two models). Hence, we expect this type of loss to reduce the complexity of the problem and to simplify the optimization process.
>
> As we discussed in Section 2.4, when the modules are properly aligned, two modules predict the same function and fulfill the same constraints. In particular, if the alignment loss (either SOB or DERL) is close to zero, module 1 and 2 output functions that are very close to each other in the Sobolev space $W^{1,2}$. As an example, we use the one in Section 2.3 on the Navier-Stokes equations. After training, module 1 has learned the momentum equation, while module 2 has learned the divergence equation. Since $u_{\text{module 1}} \approx u_{\text{module 2}}$ and $Du_{\text{module 1}} \approx Du_{\text{module 2}}$. If we calculate the divergence of module 1, we can say that $\nabla\cdot u_{\text{module 1}} \approx \nabla \cdot u_{\text{module 2}}\approx 0$. This means that a correct alignment allows module 1 to satisfy the divergence equation because module 2 does so. This intuitively provides a reason why module 1 is still capable of learning the system while seeing only one equation. We added these discussions to the revised version of the paper under Appendix G.
>
> Finally, we experimentally validated our claims that using an alignment loss instead of multiple PDE losses in a module improves the optimization and stabilizes the training procedure. In Section 3.4, we analyzed how the gradients of the different losses are backpropagated in the layers of the network. The results are clear: compared to a PINN, the gradients of the different loss terms are much more evenly distributed. This allows the modules to learn the constraints faster and more stably. The different nature of the alignment loss lets module 1 focus on a single equation, while the other is being transferred through the alignment with module 2.
> We are happy to further discuss possible directions towards a theoretical validation with the reviewer.
>
> [1] Krishnapriyan et al. Characterizing possible failure modes in physics-informed neural networks. NeurIPS 2021
> [2] Shin et al., On the convergence of physics-informed neural networks for linear second-order elliptic and parabolic type PDEs. Commun. Comput. Phys., 2021
> [3] Wang et al., When and why PINNs fail to train: A neural tangent kernel perspective. J. Comput. Phys., 2022.

---

> > ### Author Response · Authors · 2025-11-19
> > **Rebuttal (part 2/2)**
> >
> > ## W2: Conflicts between modules.
> > We thank the reviewer for raising the comment on possible conflicts between modules, as this allows us to further show the strengths of the alignment mechanism, one of our main contributions.
> >
> > The alignment mechanism we propose is specifically designed to tackle possible conflicts between the different modules in ComPhy. From a theoretical point of view, which we intuitively discussed in Section 2.4, if the models are well aligned with the SOB or DERL loss, i.e, if the alignment loss is close to zero, then the two models represent the same function in the Sobolev space $W^{1,2}$. This means that they practically represent the same function, and both can be used to predict the shared variables.
> >
> > To provide further evidence of the effectiveness of our alignment, we performed an additional test on the Kovasznay Flow and Taylor-Green experiments in Section 3. This experiment shows that the alignment makes the two aligned models functionally similar with respect to the aligned variable. At the end of training, we predict the solution for the variable $u$, which is a vector containing the x and y components of the fluid velocity, shared among the two modules. We measure the following quantities:
> > - The $L^2$ distance between the two modules, normalized with respect to the true solution norm $\frac{|| u_{\text{module 1}}-u_{\text{module 2}} ||} { || u_{\text{true}}|| }$: this indicates how different are the two predicted solutions, scaled to the magnitude of the true function.
> > - For a more point-wise comparison, we consider the symmetrized Percentage Error: for each point in the domain, we calculate $\frac{2|u_{\text{module 1}}(t,x) - u_{\text{module 2}}(t,x)|}{(|u_{\text{module 1}}(t,x)| + |u_{\text{module 2}}(t,x)|}$ which measures how far are the two predictions compared to each other. We then calculate the mean and maximum value.
> > Results on the Kovasznay flow experiment and the two combinations of ComPhy models are as follows.
> >
> > | **Model** | **Normalized $L^2$ distance** | **Mean sPE** | **Maximum sPE** |
> > |---|:---:|:---:|:---:|
> > |  | $\times 10^{-4}$ | $\times 10^{-7}$ | $\times 10^{-6}$ |
> > | CP-2xPINN (DERL+Grad) | 2.758 | 6.095 | 2.954 |
> > | CP-PINN+NCL(DERL+Grad) | 2.210 | 4.874 | 1.729 |
> >
> > While the results for the Taylor Green vortex are:
> > | **Model** | **Normalized $L^2$ distance** | **Mean sPE** | **Maximum sPE** |
> > |---|:---:|:---:|:---:|
> > |  | $\times 10^{-4}$ | $\times 10^{-7}$ | $\times 10^{-6}$ |
> > | CP-2xPINN (DERL+Grad) | 8.910 | 2.762 | 2.102 |
> > | CP-PINN+NCL(DERL+Grad) | 10.47 | 3.187 | 4.693 |
> >
> > These metrics indicate that the difference between the two modules is negligible compared to the scale of the solution or to the respective scales. We have not observed any cases where the modules disagree with each other. We added these results to our revised paper in Appendix E.
> >
> > ## W3: Efficiency.
> > We discussed the efficiency of our models in Section 2.4. We include here a more detailed discussion to show the advantages of ComPhy, which we may not have completely clarified in the paper, especially in practical terms. Each module of ComPhy is related to a single equation in the PDE system, which corresponds to a single PDE residual term in the loss. Additionally, each module has one loss term for the initial condition and one for the boundary condition. Finally, there are the alignment terms: in our experiments, as detailed at the end of section 2.1, we found it was sufficient to include one alignment loss term for each module. Overall, each ComPhy module has at most 4 loss terms. Instead, a PINN model has N+2 loss terms. For N>2, a single ComPhy module will have fewer loss terms than a PINN and thus solve a simpler optimization problem than a single PINN trained on the entire system. We also empirically show this point with our gradient analysis in Section 3.4. We remark on the same point by considering the different nature of the loss terms in our answer to W1, above.
> >
> > Finally, we highlight an important point for the practical application of ComPhy: most real-world physical systems are described by systems of 3 to 4 equations. Even if ComPhy scaled quadratically with the number of equations (it does not), the cost would remain manageable.
> >
> > ## W4: Neural Operators
> > We remark that our experiments are all unsupervised learning tasks, where no data is provided, and the models have to learn the solution solely from the definition of the PDE problem. On the other hand, Neural Operators are supervised models that require data to learn a map from the initial condition to the solution. Hence, these are two very different settings.
> > Furthermore, our approach works by separating the PDE residual losses of PINNs into different modules, but these losses are not used in training most Neural Operator models.
> >
> > We still think modular approaches like ComPhy could be applied to the Neural Operator setting, but this goes beyond the scope of our current work.

---

> ### Comment · Reviewer_AyTi · 2025-11-21
>
> Thank the authors for providing comprehensive responses, which help me understand the manuscript better. As for weakness 1, what I would expect is how your method relate to numerical methods which are often required to tackle more complicated systems of PDEs. For weaknesses 2 and 3, my concerns are solved. For weaknesses 4, I understand that the method proposed in this paper does not apply to neural operators.
> Overall, I think this is an interesting paper and I will consider raise my score later.

---

> ### Author Response · Authors · 2025-11-21
>
> We thank the reviewer for continuing to engage in this discussion. We are very happy the reviewer found our answers to be helpful and comprehensive, and that they will consider an increase in the score.
>
> To answer the outstanding question, we provide a deeper understanding of how our methodology relates to numerical methods.
>
> **Recap on why ComPhy works**
>
> ComPhy is an unsupervised learning method for PDEs, similar to PINNs and NCL, which constitute our main baselines. The way the solution to a PDE is learned is by imposing the constraints of the PDE problem (Initial Condition, Boundary condition, and PDEs) as constraints to the MLPs themselves. This is done by calculating the predictions of the model for the relevant quantities (IC, BC, or partial derivatives via automatic differentiation), calculating the PDE constraints on them, and then updating the model’s parameters based on the MSE loss. If we consider the model a real-valued function $(t,x)\rightarrow \hat{u}(t,x)$, when the losses approach zero, then the function accurately represents the solution of the PDE. Convergence guarantees are provided for some cases [1], but there are known optimization problems for the standard PINNs, as we explain in our related works section. Our model solves these issues by adopting a modular approach, where each module is related to a specific PDE, while the alignment ensures the exchange of information.
>
> **Advantages compared to numerical methods**:
>
> - Numerical methods perform their computation on a pre-defined grid of points. Since ComPhy is unsupervised, we can arbitrarily sample any point from the PDE domain for training. Therefore, we can ensure that the PDE constraints are satisfied almost everywhere.
> - During inference, ComPhy can predict the solution for any point in the domain without additional interpolation or further training, as properly trained modules should generalize well. In contrast, numerical methods calculate the solution only on the aforementioned pre-defined grid. In other points of the domain, one can either recalculate the whole solution (which can be slow, as we will show below) or use interpolation, which limits the effectiveness of the model.
> - ComPhy is very fast at inference time, especially compared to interpolation methods (see the table below).
>
> The main advantages of numerical methods against learning-based approaches like ComPhy are i) availability of theoretical guarantees on convergence, ii) a better accuracy on a fixed grid of pre-defined points with high resolution. However, the time required to calculate a solution grows very fast with increasing grid refinements. The main issue comes from the instabilities of using empirical derivatives to calculate updates, which require the time interval of the autoregressive update to be very small (see discussions on the Courant number in [2], which defines the relation between the $dt$ and $dx$ depending on the PDE order and other quantities).
>
> **Some empirical observations**
>
> To better support our claims, we ran experiments with the finite volume method on the MHD task with a high-resolution grid of points (1024x1024 points in space and 50 points in time). This will be our “ground truth” for the rest of the experiment. We then computed the finite volume solution at lower resolutions (64x64, 128x128, 256x256, and 512x512) to see the impact on the computational time and the predictive error. Finally, we calculate the error with respect to the ground truth for a PINN, NCL, and ComPhy. We also report the inference time to predict the solution on the high-resolution grid. We remark that the reported time for PINN, NCL, and ComPhy does not include training time. However, in practice, this would be the cost paid for inference (training only happens once).
>
> | **Classical Models** |  |  |
> |:---:|:---:|:---:|
> | **Resolution (space)** | **$L^2$ error** | **Computation Time / Inference Time** |
> |  | $\times 10^{-5}$ | seconds |
> | 64x64 | 16.64 | 4.40 |
> | 128x128 | 4.891 | 8.45 |
> | 256x256 | 1.342 | 50.78 |
> | 512x512 | 0.2824 | 431.7 |
> | 1024x1024 (reference solution) | — | 3853 |
> | **Machine learning Models** |  |  |
> | PINN | 1.704 | 3.88 |
> | NCL | 1.714 | 13.06 |
> | CP-2xPINN (DERL) | 1.458 | 3.46 |
> | CP-3xPINN (DERL) | 1.395 | 3.37 |
> | CP-4xPINN (DERL) | 1.423 | 3.41 |
>
> Clearly, the computational time of classical methods scales very badly with the resolution, with limited improvement in the error. We remark that the errors for the numerical methods are calculated at their own lower resolution, while machine learning models can be evaluated at 1024x1024. We conclude ComPhy is much faster during inference, has a comparable error to a high-res solution, and is the best among machine learning methods.
>
> [1] Shin et al., On the convergence of physics-informed neural networks for linear second-order elliptic and parabolic type PDEs. Commun. Comput. Phys.,
> [2] Joel H Ferziger and Milovan Peric. Computational methods for fluid dynamics. Springer, 2001.

---

### Official Review · Reviewer_amcN · 2025-10-31

**Soundness:** 3
**Presentation:** 3
**Contribution:** 3
**Rating:** 6
**Confidence:** 3

**Summary:**

This paper introduces ComPhy (CP), a modular framework for solving systems of Partial Differential Equations (PDEs) using machine learning. The core innovation involves assigning each PDE in a system to a dedicated learning module (such as PINNs or Neural Conservation Laws) and connecting these modules through an end-to-end alignment mechanism. The alignment process enforces consistency between modules that predict the same physical variables, with particular emphasis on derivative-based alignment losses (Sobolev norm and derivative-only alignment). The authors demonstrate that this compositional approach outperforms standard methods where a single model learns all PDEs simultaneously. Experiments span systems ranging from two to five equations, including Navier-Stokes, acoustic wave equations, and magnetohydrodynamics, showing consistent improvements in accuracy. The paper also provides gradient analysis suggesting that CP's modular structure leads to more balanced gradient distributions during training, which may explain its superior performance.

**Strengths:**

Strengths

The paper demonstrates several notable strengths that support its contribution to the community.
Clear motivation and intuitive approach: The paper effectively motivates the problem of solving coupled PDE systems and presents an intuitive solution. The compositional structure mirrors the mathematical structure of the underlying physical system, making the approach both theoretically appealing and practically sensible. The gradual build-up from methodology to the concrete Navier-Stokes example in Section 2.3 aids understanding.
Strong and consistent empirical results: The experimental evaluation is comprehensive, covering multiple physical systems with increasing complexity.

Thorough experimental methodology: The authors compare against multiple relevant baselines including PINN with gradient reweighting and adaptive point resampling. The experiments are well-documented with detailed problem setups, boundary conditions, and reference solutions in the appendices.

Valuable gradient analysis: Section 3.4's gradient histogram analysis provides meaningful insight into why the modular approach succeeds. The observation that CP produces more balanced gradient distributions across layers compared to standard PINNs offers an empirical explanation for the performance gains and could inform future research.
Generalization beyond divergence-free equations: Unlike NCL which is specifically designed for divergence-free fields, ComPhy's framework applies to general PDE systems, demonstrating particular value in experiments like acoustics and MHD where NCL-only approaches would be insufficient.

**Weaknesses:**

Weaknesses and Concerns

 Insufficient analysis of hyperparameter selection and sensitivity
The paper does not provide clear guidance on choosing the critical hyperparameter λ_align. While Table 2 and Table 3 show results with fixed hyperparameters, there is no ablation study examining sensitivity to this choice or methodology for setting it.

 Module assignment strategy not systematically addressed
The paper does not provide principled guidance on how to assign PDEs to modules. For the NS-Euler experiment (Section 4.1), the authors test multiple configurations (2xPINN, PINN+NCL, 2xNCL, 3xPINN) but offer no systematic approach for making this choice. Different assignments can lead to different architectures, but the selection process appears ad-hoc.
The paper would benefit from either developing heuristics for module assignment (e.g., based on PDE type, coupling strength, or variable sharing patterns) or demonstrating that the method is robust across reasonable assignment choices.

 Comparison fairness and architecture choices
The baseline single PINN appears to use the same architecture size as individual CP modules (Table 5), meaning the total CP model has substantially more parameters across all modules. It is unclear whether a larger single PINN with comparable total parameter count would close the performance gap. I wonder whether the observed gains arise specifically from the modular structure or merely from the increased model capacity.

**Questions:**

Statistical significance and error bars
The results tables (Tables 2 and 3) report point estimates without error bars or confidence intervals. Given that neural network training involves stochastic elements (random initialization, batch sampling), reporting means and standard deviations across multiple runs would strengthen the claims.

Ablation on alignment losses
The authors should include ablation studies showing performance across a range of λ_align values, demonstrate the method's robustness (or lack thereof) to hyperparameter choices

Computational Efficiency and Cost–Performance Analysis
The training time overhead of CP models is mentioned but not quantitatively justified. It would be useful to discuss whether the additional training cost is proportionate to the observed performance improvement. Presenting these metrics side by side would improve the completeness and transparency of the experimental analysis.

Parameter-matched baseline
Compare against a single PINN baseline that is parameter-matched to the full CP system (same total parameter count). Alternatively, show scaling curves where single PINN and CP models are compared across increasing total parameter budgets. This will clarify whether gains are due to modularity or simply model capacity.

Alignment learning
Recent studies [1][2] have incorporated alignment learning into PINN-like frameworks, demonstrating its potential to enhance physical consistency and optimization efficiency. Therefore, the authors should discuss how their proposed method relates to these works, highlighting the key differences or advantages.

[1]Gradient Alignment in Physics-informed Neural Networks: A Second-Order Optimization Perspective

[2]Physics-informed Temporal Alignment for Auto-regressive PDE Foundation Models

---

> ### Author Response · Authors · 2025-11-19
> **Rebuttal (part 1/2)**
>
> We thank the reviewer for their comments and positive feedback on our clarity, experiments, and gradient analysis.
>
> We now address the weaknesses and specific questions raised.
> First, we would like to remark that our experiments follow a rigorous experimental design, with reproducibility guarantees via fixed seed and a clear hyperparameter selection process. Getting rid of hyperparameters entirely is unlikely, as hyperparameter selection is part of any PINN optimization, and of any machine learning procedure in general.
> ## W1: Hyperparameter sensitivity.
> We understand the reviewer's concern about providing convincing proof of robustness with respect to the hyperparameter search and the choice of the $\lambda_{\text{align}}$. We would like to remark that in the test cases in Section 3,  the ComPhy models without gradient-based reweighting are trained with the naive choice $\lambda_{\text{align}}=1$, hence showing the performance of ComPhy “out of the box”. Nevertheless, our best models in the case studies are those with gradient-based reweighting, which are not affected by the burden of hyperparameter selection and hence work without carefully selecting the weights.
> For the rebuttal, we performed some additional experiments and ablations by varying the value of $\lambda_{\text{align}}$ to show the sensitivity of ComPhy with respect to this parameter on the acoustic equations experiment.
> | **Acoustic equations** |  |  |
> |:---:|:---:|:---:|
> | **Model** | **$L^2$ error** | **max_err** |
> | CP-3xNCL (DERL) | $\times 10^{-5}$ | $\times 10^{-1}$ |
> | $\lambda_{\text{align}}=1$ | 5.165 | 1.547 |
> | $\lambda_{\text{align}}=0.1$ | 3.164 | 1.257 |
> | $\lambda_{\text{align}}=0.01$ | 2.460 | 1.089 |
> | $\lambda_{\text{align}}=10.$ | 12.46 | 4.388 |
>
> As we can see, $\lambda_{\text{align}}$ regulates the importance of the alignment between models. The $\lambda_{\text{align}}=1$ works well in our experiments, while highly specialised tuning of the hyperparameter can lead to modest improvement, as in this.
> We added these results to our ablations in Appendix F.
>
> ## W2: module assignment strategy
> We agree with the reviewer that the choice of the modules is relevant for the best performance of the overall model. In our experiments, apart from the MHD one, where we tested the scaling capabilities of ComPhy, we explored various combinations to demonstrate the flexibility and robustness of ComPhy to different choices of learning modules. Depending on the system of PDEs, we tried every possible combination of modules, given that the NCL module requires a divergence-free equation.
>
> Our results support the fact that, when possible, a combination of modules that includes NCL is always beneficial. We will make this point clearer in the updated version of the paper. From a practical point of view, we suggest using NCL modules when there is one or more divergence-free equations. Similarly, since most physical systems present 2 or 3 equations, the most common and effective choice is to use 2 or 3 modules, respectively.
>
> We add a similar discussion at the end of Section 4.
>
> ## W3: on the comparison fairness
> We understand the need for a fair empirical comparison between a single PINN and a ComPhy model with multiple PINN modules. First, we would like to stress that, during inference, ComPhy uses only one module to predict the solution, and, in practical applications, we would expect the training phase to be performed once, while inference is performed multiple times. Given the fact that these models are not prohibitively large in the number of parameters, doubling the size of the model does not make a huge difference during training, while the cost during inference could accumulate. That is the reason why we preferred using the same number of parameters at inference instead of at training time.
>
> Nevertheless, we made some additional experiments on this topic by varying the number of units in the MLP layers in the baselines to match the parameter budget as the whole ComPhy model (that is, the sum of parameters in the modules). The results are in the following table, where we consider our best CP models and the PINN and NCL baselines for the Acoustic equation experiments.
> | **Acoustic equations** |  |  |
> |:---:|:---:|:---:|
> | **Model** | **$L^2$ error** | **max_err** |
> |  | $\times 10^{-5}$ | $\times 10^{-1}$ |
> | **PINN** | 5.671 | 1.462 |
> | **NCL** | 2.940 | 1.169 |
> | **CP-3xNCL (DERL+Grad)** | **2.718** | **1.121** |
>
> As we can see, our models perform better even though they use fewer parameters during inference, further showing the importance of modularization in these applications.
> We added these results to our ablations in Appendix F.

---

> > ### Author Response · Authors · 2025-11-19
> > **Rebuttal (part 2/2)**
> >
> > ## Q1: statistical significance.
> >
> > We followed the common practices of the literature on PINNs, where statistical significance and error bars are not included. Furthermore, our experiments are completely reproducible, as we provide code and a fixed seed for training, which was not cherry-picked.
> >
> > We performed additional experiments on PINN, NCL, and a ComPhy model on the acoustic equation with different seeds to show the robustness of our method to different initializations and training stochasticity. The following table reports the mean and standard deviation for the relevant metrics across 3 different seeds.
> > | **Acoustic equations** |  |  |
> > |:---:|:---:|:---:|
> > | **Model** | **$L^2$ error** | **max_err** |
> > |  | $\times 10^{-5}$ | $\times 10^{-1}$ |
> > | PINN | 5.090$_{\pm0.859}$ | 2.247$_{\pm0.413}$ |
> > | NCL | 3.540$_{\pm0.431}$ | 1.610$_{\pm0.082}$ |
> > | CP-3xNCL (DERL) | 1.723$_{\pm0.173}$ | 1.091$_{\pm0.088}$ |
> >
> > As we can see, the results clearly indicate that the CP model with 3 NCL modules is still the best model by far, with lower $L^2$ and maximum errors in the domain. Furthermore, the standard deviations are smaller, indicating more robustness to different initializations. We added these results to our ablations in Appendix F.
> >
> > ## Q2: ablations
> >
> > See our previous comments on W1.
> >
> > ## Q3 and Q4: computational efficiency.
> > We performed additional experiments where PINN and NCL models have the same parameter budget as a ComPhy model during training, which leads to 2 or 3 times more during inference compared to ComPhy. Results indicate no relevant performance improvement (see W3). We think this shows that the initial additional computational cost is worth the performance improvement during inference. We added these results to our ablations in Appendix F.
> >
> > ## Q5:
> > We thank the reviewer for pointing out interesting works in the field of physics-informed models and alignment. We include and discuss these works in the revised manuscript in our related works Section.
> >
> > [1] focuses on the alignment between gradients in the gradient descent, employing and analysing different (quasi-)second-order corrections for gradient updates for faster and improved convergence when PINN objectives are contrasting. While these approaches can improve the training of PINNs, second-order corrections are often computationally costly, scaling worse than linearly.
> >
> > [2] is based on autoregressive and data-driven models (such as Neural Operators), which are very different from our setting, as our model and baselines are unsupervised and work without autoregressive rollouts. They also consider equation discovery in their pipeline and impose that the equation fitted on the available data is aligned with the ground truth equation as an additional constraint.
> >
> > References:
> >
> > [1] Gradient Alignment in Physics-informed Neural Networks: A Second-Order Optimization Perspective
> >
> > [2] Physics-informed Temporal Alignment for Auto-regressive PDE Foundation Models

---

> > > ### Comment · Reviewer_amcN · 2025-11-24
> > > **Comment**
> > >
> > > Thank you for the detailed response, which has addressed all of my concerns. I will maintain my positive rating.

---

> > > > ### Author Response · Authors · 2025-11-25
> > > >
> > > > We thank the reviewer for acknowledging our effort during the rebuttal. Given the positive feedback in the original review and our rebuttal, which has addressed all their concerns, we would kindly ask the reviewer to consider raising the score if there is no outstanding issue or question they would like us to address further, to reach a fully positive score of our work consistent with their comments.

---

> > > > > ### Comment · Reviewer_amcN · 2025-11-26
> > > > >
> > > > > Considering the method's potential future impact on the field, as well as the concerns raised by other reviewers, I believe maintaining the score of 6 is reasonable.

---

### Author Response · Authors · 2025-11-19
**Rebuttal Summary**

We thank the reviewers for their time and for providing valuable feedback on our work. Many positive comments were posted by the reviewers, highlighting the ***clear motivation and intuitive approach*** of our work (reviewer amcN). Our modular approach has been received as being ***novel*** (revs. Jj5v, AyTi, M37D) and ***elegant for solving systems of PDEs*** (revs. Jj5v, AyTi), with our Sobolev-based alignment that ***effectively transfers physical information between modules and leads to empirical gains*** (rev. Jj5v). We are happy that our efforts to provide empirical evidence of the advantages have been appreciated: our gradient analysis ***provides meaningful insight*** (rev. amcN) and ***convincingly explains why ComPhy’s modular approach stabilizes training compared to conventional PINNs*** (rev. Jj5v). Our experimental validation is ***thorough and well-detailed*** (rev. amCN) with ***rich implementation details*** (reviewer M37D) and delivers a comprehensive evaluation of the proposed method (rev. M37D)

We address the specific comments by the reviewers in the corresponding threads. Here, we highlight the main additions of our rebuttal, which we included in the revised manuscript of the paper with a blue color to highlight the differences:
- Ablations on $\lambda_{\text{align}}$, showing no drastic change in performance with modest changes in its value (revs. amcN and Jj5v).
- Practical considerations to decide the best combination of modules, based on the system of PDEs (revs. amcN and M37D).
- Ablations on the number of parameters in the PINN and NCL models, showing that our model still performs better than the baselines even with fewer parameters during inference (revs. amcN and AyTi).
- Experiments with multiple runs, showing that our results have statistical significance (rev. amcN).
- We verified that modules do not provide conflicting predictions. By measuring relative distances and symmetric percentage errors between the modules’ predictions, we further showed the strength of the alignment approach (rev. AyTi).
- Further clarifications on the advantages of ComPhy, focusing on the differences between a PDE residual term and the alignment loss, which facilitates training, as demonstrated by our gradient analysis (revs. AyTi and M37D).
- Loss curves for ComPhy models, showing the correlation between alignment, the transfer of physical constraints, and the prediction loss (rev. M37D).

---

### Author Response · Authors · 2025-11-28
**Wrap-up**

We are delighted that our discussion with the reviewers during the rebuttal process demonstrated the value and impact of our work. As summarized in our discussion below, all concerns were addressed and solved, with the reviewers being satisfied by our answers and ultimately leaning towards acceptance.

---

### Author Response · Authors · 2025-11-29
**Discussion Phase Summary**

Given the decision of the ICLR Conference chairs to revert the scores to their pre-rebuttal values and given that the discussion phase no longer continues as originally planned, we hereby provide a summary of the rebuttal process to the new Area Chair of this paper.
In particular, all the concerns raised by the reviewers have been addressed properly, and we were able to answer any follow up questions. In the end:
- Reviewer **amcN** was satisfied with our rebuttal, as we *addressed all their concerns*, confirming to lean towards acceptance. However, despite the very positive feedback, they did not increase the score and, when asked, did not provide further actionable items or specific questions to address.
- Reviewer **AyTi** was *considering a raise in the score* and posted a follow-up question, to which we provided an in-depth answer with additional results.
- Reviewer **Jj5v** was already leaning towards acceptance, highlighting our *elegant and well-motivated* approach and our analysis on gradient distribution. In our rebuttal, we addressed his concerns with ablations and further analysis, but the reviewer never responded.
- Reviewer **M37D** was very enthusiastic about our *detailed* answers, as they increased their vote from 2 to 6, which we find impressive. They also posted a follow-up question, to which we answered promptly with new insights on the alignment mechanism, which we think strengthens our results.

A summary of the changes and answers, which we already implemented in the revised manuscript, is provided in an earlier comment.
We understand the additional burden from this modified workflow for the decision, and we kindly ask the Area Chair to consider our comprehensive responses and additional results, which we think further improve our work.

Best regards,
The authors

---

### Meta-Review · Area_Chair_Dmfx · 2026-01-06

**Summary:**

This paper proposes ComPhy, a modular framework for solving systems of PDEs. The key idea is to assign each equation a dedicated learning module, like PINN or NCL, and introduce an end-to-end alignment mechanism to enable consistency between different modules. Reviewers generally view the approach as well motivated, novel, and elegant, particularly appreciating the modular design and the alignment strategy, which provides both conceptual clarity and empirical benefits. Initial concerns focused on hyperparameter sensitivity, fairness of comparisons, training stability, and the robustness of empirical gains. Overall, the rebuttal strengthens the paper by addressing these concerns with additional analysis and experiments, leading to a more convincing case for the effectiveness and practicality of the proposed method.

**Reviewer Concerns:**

The rebuttal addresses the majority of reviewer concerns in a satisfactory manner. The authors provide ablation studies demonstrating limited sensitivity to key hyperparameters, clarify practical guidelines for selecting module combinations, and include parameter-count ablations showing that ComPhy outperforms baselines even with fewer parameters at inference time. Additional experiments with multiple runs establish statistical significance, addressing concerns about robustness.

The authors also strengthen the methodological justification by verifying that different modules do not produce conflicting predictions and by providing deeper analysis of the alignment loss, highlighting how it differs from standard PDE residual terms and contributes to more stable training. Loss curves and gradient analyses further clarify the relationship between alignment and improved optimization behavior.

**Reviewer Scores:**

Reviewer amcN has indicated to maintin the positive score 6.

Reviewer AyTi may raise the score from 4 to 6 since most concerns are adequately addressed.

Reviewer Jj5v may keep the score 6 or increase to 8 given that the authors provided additional experments to clarify concerns about computational efficiency and hyperparameters.

Reviewer M37D has indicated to increase the score from 2 to 6.

---

### Decision · Program_Chairs · 2026-01-26

Accept (Poster)